statistics/applied mathematics/ mathematical modelling

posterior-based proposal, Markov chain Monte Carlo, Bayesian inference, mixed model, stochastic volatility, statistical genetics

**Author for correspondence:**
C. M. Pooley
e-mail: christopher.pooley@roslin.ed.ac.uk

†Deceased.

# Posterior-based proposals for speeding up Markov chain Monte Carlo

C. M. Pooley[1,2], S. C. Bishop[1,†], A. Doeschl-Wilson[1] and G. Marion[2]

[1]The Roslin Institute, The University of Edinburgh, Midlothian EH25 9RG, UK
[2]Biomathematics and Statistics Scotland, James Clerk Maxwell Building, The King's Buildings, Peter Guthrie Tait Road, Edinburgh EH9 3FD, UK

CMP, 0000-0002-8779-4477

Markov chain Monte Carlo (MCMC) is widely used for Bayesian inference in models of complex systems. Performance, however, is often unsatisfactory in models with many latent variables due to so-called poor mixing, necessitating the development of application-specific implementations. This paper introduces 'posterior-based proposals' (PBPs), a new type of MCMC update applicable to a huge class of statistical models (whose conditional dependence structures are represented by directed acyclic graphs). PBPs generate large joint updates in parameter and latent variable space, while retaining good acceptance rates (typically 33%). Evaluation against other approaches (from standard Gibbs/random walk updates to state-of-the-art Hamiltonian and particle MCMC methods) was carried out for widely varying model types: an individual-based model for disease diagnostic test data, a financial stochastic volatility model, a mixed model used in statistical genetics and a population model used in ecology. While different methods worked better or worse in different scenarios, PBPs were found to be either near to the fastest or significantly faster than the next best approach (by up to a factor of 10). PBPs, therefore, represent an additional general purpose technique that can be usefully applied in a wide variety of contexts.

## 1. Introduction

Markov chain Monte Carlo (MCMC) techniques allow correlated samples to be drawn from essentially any probability distribution by iteratively generating successive values of a carefully constructed Markov chain. This flexibility has led MCMC to become the method of choice for inferring model parameters under Bayesian inference [1]. However, for high-dimensional systems (e.g. where inference is over many tens, hundreds or even thousands of variables), MCMC often suffers from a problem

known as 'poor mixing'. This manifests itself as a high degree of correlation between consecutive samples along the Markov chain, so requiring a very large number of iterations to adequately explore the posterior [1]. This limitation is of practical importance, because it restricts the possible models to which MCMC can realistically be applied. The focus of this paper is to introduce and explore a new approach that helps alleviate these mixing problems, thus reducing the computational time necessary to generate accurate inference. This approach has practical advantages over existing methodologies that aim to address the same problem [2–4].

In Bayesian inference, the posterior distribution $\pi(\theta, \xi | y)$ represents the state of knowledge concerning the parameters $\theta$ and latent variables $\xi$ of a given stochastic model taking into account data $y$. Using Bayes' theorem, this posterior distribution can be expressed as

$$\pi(\theta, \xi | y) = \frac{\pi(y | \xi, \theta) \pi(\xi | \theta) \pi(\theta)}{\pi(y)}, \tag{1.1}$$

where $\pi(y | \xi, \theta)$ is here referred to as the observation model, $\pi(\xi | \theta)$ is the latent process (i.e. the part of the model which is not directly observed but helps explain the observations) likelihood, $\pi(\theta)$ is the prior distribution (representing the state of knowledge prior to data $y$ being considered) and $\pi(y)$ is a constant factor known as the model evidence [5].

An MCMC implementation of Bayesian inference aims to produce samples from the posterior, i.e. a list of parameter values $\theta^i$ and latent variables $\xi^i$ distributed in accordance with equation (1.1). This is achieved by sequentially proposing some change to the current state $\theta^i$ and/or $\xi^i$ to generate $\theta^p$ and $\xi^p$, and accepting or rejecting this change with a Metropolis–Hastings (MH) probability to create the next member on the list, i.e. $\theta^{i+1}$ and $\xi^{i+1}$ [6]. Note, in some instances, it is possible to probabilistically generate samples directly, e.g. via Gibbs sampling [7] or slice sampling [8], without the need for an accept/reject step.[1] The term Monte Carlo refers to the probabilistic nature of these updates that form a Markov chain, i.e. step $i + 1$ in the chain only depends on the state at the previous step.

A typical approach is to sequentially update each parameter and latent variable separately. Figure 1$a$ illustrates the problem of poor mixing resulting from implementing this 'standard' MCMC when there is a strong dependency between latent variables and model parameters. The dark shaded area represents high posterior probability[2] as a function of $\theta$ and $\xi$. Consider first fixing $\theta$ and making changes to $\xi$, as shown by the dashed line in figure 1$a$. Because MCMC samples are probabilistically constrained to lie in the shaded region, the chain will make limited progress even for a large number of MH updates. Similarly, changes to $\theta$ while fixing $\xi$ will be restricted to move along horizontal lines, which are again limited in scope. A typical output from an MCMC algorithm which independently updates parameters and latent variable is illustrated in figure 1$b$, which shows the trace plot for one of the variables in $\theta$. Thousands of MH updates are potentially needed to generate just one uncorrelated effective sample from the posterior.

The way out of this sorry state of affairs is shown in figure 1$c$. Here the proposals are performed *jointly* in parameter and latent variable space (as indicated by the pink shaded region) and share a similar correlated structure to the posterior itself. In this case, proposals can jump much further without the posterior probability becoming negligibly small. Consequently, as illustrated in figure 1$d$, successive samples are less correlated and fewer are needed to be representative of the posterior.

Various techniques to perform joint updates have been proposed in the literature. For example particle MCMC (PMCMC) [2] samples a new set of parameters $\theta^p$ relative to $\theta^i$ and then sets about generating $\xi^p$. This is achieved by directly sampling from the model $\pi(\xi | \theta^p)$ multiple times, with each instance referred to as a 'particle'. The final result is built up in a series of stages that sequentially take into account a larger fraction of the data $y$. At the end of each stage, those particles which agree well with the sequentially introduced observations are duplicated at the expense of those which don't agree so well (so-called particle filtering). While this method exhibits very good mixing, computational speed is often compromised because the number of particles needed to generate a reasonable acceptance probability can be very large [9].[3]

Approximate Bayesian computation (ABC) [3] also samples directly from the model, but rather than fitting the data to the samples through a full observation model, as in equation (1.1), here the fit with the data is characterized by a much simpler distance measure $\chi$. Often $\chi$ is specifically chosen such that a substantial proportion of simulated samples contribute to the final result. Such an approach, however, comes at a

---

[1] In these cases, the MH acceptance probability is exactly one.

[2] Note, this diagram is purely schematic, as $\theta$ and $\xi$ are usually multidimensional quantities.

[3] Potentially leading to detrimentally large computational memory requirements.

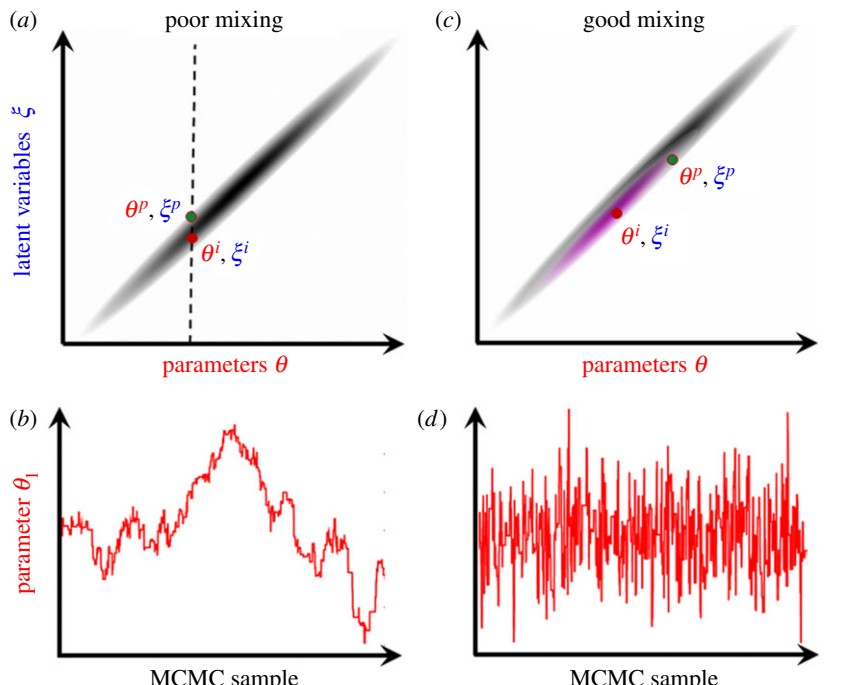

**Figure 1.** Mixing. Illustrative example of poor/good mixing. (*a*) Proposals are made individually on parameters and latent variables separately (the dashed line shows the case of a latent variable being changed). The shading represents the region of high posterior probability. (*b*) Trace plot exhibiting poor mixing. (*c*) Efficient joint proposals are made using the distribution in pink (which is correlated in the same way as the posterior). (*d*) Trace plot exhibiting good mixing.

significant cost, because ABC generates only approximate, rather than exact, draws from the posterior [10]. Furthermore, doubts have been cast on ABC's ability to accurately perform model selection [10,11].

In Hamiltonian MCMC (HMCMC) [3,12] $\theta^i$ and $\xi^i$ are dynamically changed through a series of small intermediary steps (which take into account local gradients in the log of the posterior probability) to reach $\theta^p$ and $\xi^p$. This final state is then accepted or rejected with an overall MH probability. This, again, produces good mixing (due to the fact that $\theta^i, \xi^i$ and $\theta^p, \xi^p$ can be widely separated), but its efficiency is critically dependent on the number of intermediary steps needed for a sufficiently high acceptance rate [3]. Although recent improvements have helped to optimize this technique (most notably the 'No-U-turn sampler' introduced in [13]), HMCMC is applicable to only those models for which $\theta$ and $\xi$ are continuous quantities. Consequently, it is not well suited to tackle models with discrete variables, e.g. disease status [14], or variable dimension number, e.g. event data [15].

Another approach is to develop a non-centred parametrization (NCP) [4,16,17] in which a new set of latent variables $\xi'$ (whose distributions are typically independent of the model parameters $\theta$) are introduced. These new variables are related to the original latent variables through some deterministic function $\xi = h(\xi', \theta, y)$. MCMC implemented using this reparameterization can lead to improved mixing [17]. In this case, proposed changes to $\xi'$ can be thought of as joint proposals in both $\xi$ and $\theta$.

This paper introduces a new class of MCMC proposal valid for the vast majority of statistical models (§2). We refer to these as 'posterior-based proposals' (PBPs, §3), as they are constructed with the aid of importance distributions (§4) which approximate the posterior. PBPs enable joint updates to both $\theta$ and $\xi$ (unlike standard approaches), are fast (i.e. they don't require multiple particles like PMCMC), accurate (i.e. they draw samples from the true posterior, unlike ABC) and they can be applied to continuous or discrete state-space models (unlike HMCMC). A further novelty in PBPs is that they not only account for correlations between $\theta$ and $\xi$ inherent to the model, but can also take into account the data (unlike NCP). Application to models used in disciplines ranging from statistical genetics to epidemiology to finance demonstrate that PBPs potentially offer considerable improvements in performance over standard approaches (§5).

# 2. Broadly applicable model framework

PBPs are potentially applicable to any statistical model whose conditional dependence structure can be represented by a directed acyclic graph (DAG) [18], as illustrated in figure 2. This encompasses a vast

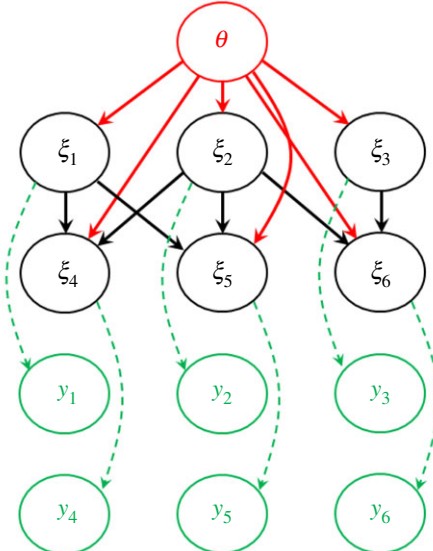

**Figure 2.** Directed acyclic graph (DAG). This shows a model with parameters $\theta$, latent variables $\xi$ and observations $y$. The arrows represent conditional dependencies. The model assumes that latent variables $\xi_e$ (where $e$ goes from 1 to $E = 6$ in this example) are sampled from a set of univariate probability distributions $\pi(\xi_e | \xi_{e' < e}, \theta)$ and $y_r$ are sampled from $\pi(y_r | \xi, \theta)$ (note, this example shows the special case when $y_r$ depends only on $\xi_r$).

range of statistical models including mixed models (MMs) [19], generalized linear MMs [20], hidden Markov models [21], discrete-time Markov processes [22] and most of the models that can be defined in automated Bayesian software, such as WinBUGS [23], JAGS [24] or Stan [25], which specifically assume a DAG structure. A key property of DAGs is that the indices for the latent variables $e$ can be ordered such that each element $\xi_e$ is conditionally dependent on only those other elements with lower index $\xi_{e' < e}$ (a property known as topological ordering).[4]

Simulation results from sequentially sampling each latent variable $\xi_e$ (starting from $e = 1$ up to $e = E$) from a set of model-defining univariate probability distributions $\pi(\xi_e | \xi_{e' < e}, \theta)$. Consequently, the latent process likelihood in equation (1.1) can be expressed as

$$\pi(\xi | \theta) = \prod_{e=1}^{E} \pi(\xi_e | \xi_{e' < e}, \theta). \tag{2.1}$$

# 3. Posterior-based proposals

## 3.1. Aim

PBPs first propose a new set of parameters $\theta^p$ relative to $\theta^i$ and then generate $\xi^p$ by means of stochastically *modifying* $\xi^i$ to account for this change in parameters (note, this is in stark contrast to PMCMC which aims to *sample* $\xi^p$ directly from $\pi(\xi | \theta^p, y)$ without reference to $\xi^i$ or $\theta^i$). The novel PBP process involves sequentially sampling each latent variable $\xi_e^p$ from $e = 1$ up to $E$, and makes use of so-called importance distributions (IDs) applied to both the initial and proposed states. These importance distributions $f_{ID}(\xi_e | \xi_{e' < e}, \theta, y)$ are approximations to the posterior distributions $\pi(\xi_e | \xi_{e' < e}, \theta, y)$ used in importance sampling (see electronic supplementary material, appendix A for further details). For clarity, we leave a discussion of how these approximations are made in practice until §4.

## 3.2. Example

We first run through an illustrative example of a PBP and then provide a general description in §3.3. Figure 3 shows hypothetical distributions for a particular value of $e$ for which $\xi_e$ takes non-negative integer values. Here, the black lines represent the true (unknown) distributions for the current

---

[4]The notation $\xi_{e'<e}$ denotes all elements in $\xi$ with index smaller than $e$.

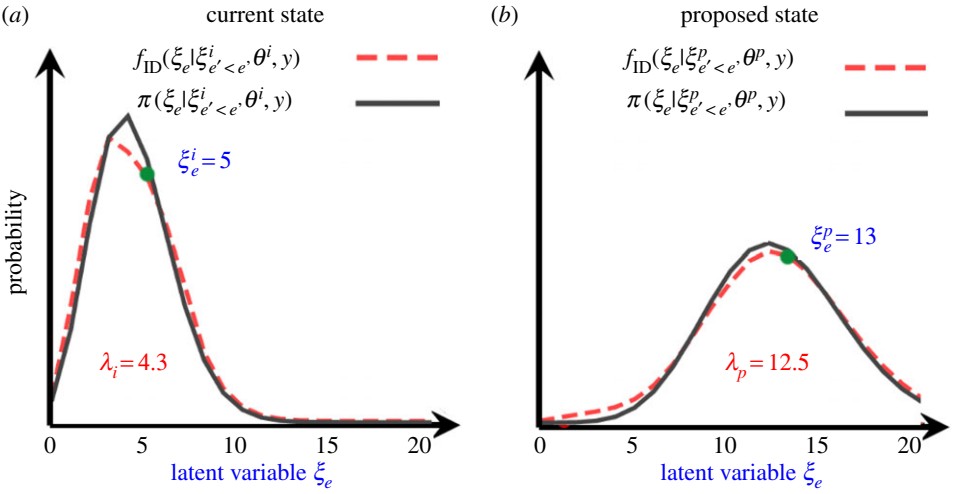

**Figure 3.** PBP updates. Shows the true (unknown) distribution $\pi(\xi_e|\xi_{e'<e},\theta,y)$ (black lines) and importance distribution $f_{ID}(\xi_e|\xi_{e'<e},\theta,y)$ (dashed red lines) for a given latent variable $\xi_e$ using (a) the current state on the MCMC chain and (b) the proposed state. In this particular example, $\xi_e$ takes non-negative integer values and the IDs are Poisson distributions.

$\pi(\xi_e|\xi^i_{e'<e},\theta^i,y)$ and proposed $\pi(\xi_e|\xi^p_{e'<e},\theta^p,y)$ states, respectively. Since these curves are approximately Poisson distributed,[5] the IDs are taken to be Poisson (as shown by the red dashed lines in figure 3) with probability mass functions (p.m.f.)

$$f_{ID}(\xi_e|\xi^i_{e'<e},\theta^i,y) = \frac{e^{-\lambda_i}\lambda_i^{\xi_e}}{\xi_e!},$$
$$f_{ID}(\xi_e|\xi^p_{e'<e},\theta^p,y) = \frac{e^{-\lambda_p}\lambda_p^{\xi_e}}{\xi_e!}. \tag{3.1}$$

These functions are characterized by 'expected event number' parameters $\lambda_i$ and $\lambda_p$, which themselves are functionally dependent on $(\xi^i_{e'<e}, \theta^i, y)$ and $(\xi^p_{e'<e}, \theta^p, y)$, respectively (see §4).

A unique feature of PBPs is that the sampling distribution for $\xi^p_e$ crucially depends on the relative size of $\lambda_p$ and $\lambda_i$. When $\lambda_p > \lambda_i$ (as it is in figure 3) $\xi^p_e$ is generated by adding a Poisson-distributed variable onto $\xi^i_e$ with expected event number given by the *difference* between $\lambda_p$ and $\lambda_i$:

$$\xi^p_e = \xi^i_e + X \quad \text{where } X \sim \text{Pois}(\lambda_p - \lambda_i), \tag{3.2}$$

(e.g. in figure 3 a random Poisson sample $X = 8$ results in $\xi^p_e = \xi^i_e + X = 13$). Such an approach makes sense, because adding an approximately Poisson distributed quantity with expected event number $\lambda_i$, to one with expected event number $\lambda_p - \lambda_i$, gives an approximately Poisson distributed variable with expected event number $\lambda_p$, as required by $\xi^p_e$ on the left-hand side of equation (3.2). Thus, equation (3.2) modifies $\xi^i_e$ to generate $\xi^p_e$ accounting for the change in $\lambda$.

On the other hand, when $\lambda_p \leq \lambda_i$ the actual number of events in the proposed state $\xi^p_e$ should be less than in the initial state $\xi^i_e$, because the expected number of events parameter $\lambda$ has reduced. Specifically, each existing event in $\xi^i_e$ is retained in $\xi^p_e$ with probability $\lambda_p/\lambda_i$, which is equivalent to sampling $\xi^p_e$ from the following binomial distribution[6]

$$\xi^p_e \sim B\left(\xi^i_e, \frac{\lambda_p}{\lambda_i}\right). \tag{3.3}$$

Together, the two potential sampling schemes for $\xi^p_e$ in equations (3.2) and (3.3) (selected depending on whether $\lambda_p$ is bigger or smaller than $\lambda_i$) make up the PBP proposal in cases in which the ID is Poisson. This is summarized by the first line in table 1.

---

[5]A Poisson distribution expresses the probability a given number of events occur in a fixed interval of time, assuming that events occur at a constant rate.

[6]If $N$ represents the total number of experiments and $p$ is the success probability of each experiment then $B(N,p)$ samples the number of successes.

**6**

**Table 1.** How to sample the proposed latent variable $\xi_e^p$ from $\xi_e^i$ (right-hand column) [not subscript] given some importance distribution (ID) (left-hand column) with functional form given by the middle column. Note the choice of proposal is typically dependent on the relative size of the characteristic parameters for the initial and proposed state. (For reference, the hypergeometric distribution $X \sim HG(N,K,n)$ has p.m.f. $C_X^K C_{n-X}^{N-K}/C_n^N$ and the negative hypergeometric distribution $X \sim NHG(N,K,r)$ has p.m.f. $C_X^{X+r-1}C_{K-X}^{N-r-X}/C_K^N$.)

| ID | prob. dist. | posterior-based proposal |
| --- | --- | --- |
| **(a)** | | |
| Poisson $\xi_e \sim Pois(\lambda)$ | $\dfrac{\lambda^{\xi_e} e^{-\lambda}}{\xi_e!}$ | $\lambda_p > \lambda_i$ : $\xi_e^p = \xi_e^i + X$ where $X \sim Pois(\lambda_p - \lambda_i)$ |
| | | $\lambda_p \leq \lambda_i$ : $\xi_e^p \sim B\left(\xi_e^i \frac{\lambda_p}{\lambda_i}\right)$ |
| normal $\xi_e \sim N(\mu,\sigma^2)$ | $\dfrac{1}{\sqrt{2\pi\sigma^2}}e^{-\frac{(\xi_e-\mu)^2}{2\sigma^2}}$ | $\sigma_p > \sigma_i$ : $\xi_e^p \sim N(\mu_p + \alpha(\xi_e^i - \mu_i), \kappa(\sigma_p^2 - \sigma_i^2))$ where $\alpha^2 = \kappa + (1-\kappa)(\sigma_p^2/\sigma_i^2)$ |
| | | $\sigma_p \leq \sigma_i$ : $\xi_e^p \sim N\left(\mu_p + \alpha\frac{\sigma_p^2}{\sigma_i^2}(\xi_e^i - \mu_i), \kappa\frac{\sigma_p^2}{\sigma_i^2}(\sigma_i^2 - \sigma_p^2)\right)$ where $\alpha^2 = \kappa + (1-\kappa)(\sigma_i^2)(\sigma_i^2/\sigma_p^2)$ |
| | | $\kappa$ is a tuneable constant (typically 0.03) |
| exponential $\xi_e \sim Exp(r)$ | $re^{-r\xi_e}$ | $r_p > r_i$ : $X \sim Exp(r_p - r_i)$, if $X > \xi_e^i$ then $\xi_e^p = X$ |
| | | else $\xi_e^p = \xi_e^i$ |
| | | $r_p \leq r_i$ : $u$ is a random number between 0 and 1 |
| | | if $u < 1 - \frac{r_p}{r_i}$ then $\xi_e^p = \xi_e^i + X$, $X \sim Exp(r_p)$ |
| | | else $\xi_e^p = \xi_e^i$ |
| Gamma $\xi_e \sim \Gamma(\alpha,\beta)$ | $\dfrac{\beta^\alpha \xi_e^{\alpha-1} e^{-\beta\xi_e}}{\Gamma(\alpha)}$ | $\alpha_p > \alpha_i$ : $\xi_e^p = \frac{\beta_i}{\beta_p}(\xi_e^i + X)$ where $X \sim \Gamma(\alpha_p - \alpha_i, \beta_i)$ |
| | | $\alpha_p \leq \alpha_i$ : $\xi_e^p = \frac{\beta_i}{\beta_p}\xi_e^i Y$ where $Y \sim Beta(\alpha_p\alpha_i - \alpha_p)$ |

(Continued.)

**Table 1.** (*Continued.*)

| ID | prob. dist. | posterior-based proposal | |
|---|---|---|---|
| Beta $\xi_e \sim Beta(\alpha, \beta)$ | $\dfrac{\Gamma(\alpha+\beta)}{\Gamma(\alpha)\Gamma(\beta)}\xi_e^{\alpha-1}(1-\xi_e)^{\beta-1}$ | $\alpha_p > \alpha_i$ $\beta_p = \beta_i$ | $\xi_e^p = 1 - (1-\xi_e^i)X$ where $X \sim Beta(\beta_i + \alpha_i, \alpha_p - \alpha_i)$ |
| | | $\alpha_p \leq \alpha_i$ $\beta_p = \beta_i$ | $\xi_e^p = \dfrac{\xi_e^i X}{1 - \xi_e^i(1-X)}$ where $X \sim Beta(\alpha_p, \alpha_i - \alpha_p)$ |
| | | $\beta_p > \beta_i$ $\alpha_p = \alpha_i$ | $\xi_e^p = \xi_e^i X$ where $X \sim Beta(\alpha_i + \beta_i, \beta_p - \beta_i)$ |
| | | $\beta_p \leq \beta_i$ $\alpha_p = \alpha_i$ | $\xi_e^p = \dfrac{\xi_e^i}{\xi_e^i + (1-\xi_e^i)X}$ where $X \sim Beta(\beta_p, \beta_i - \beta_p)$ |
| Bernoulli $\xi_e \sim Bern(z)$ | $Pr(\xi_e = 1) = z$ $Pr(\xi_e = 0) = 1 - z$ | $z_p > z_i$ | if $\xi_e^i = 1$ set $\xi_e^p = 1$ else $\left\{\text{if } u < \frac{z_p - z_i}{1 - z_i} \text{ then set } \xi_e^p = 1 \text{ else } \xi_e^p = 0\right\}$ where $u$ is a random between 0 and 1 |
| | | $z_p \leq z_i$ | if $\xi_e^i = 0$ set $\xi_e^p = 0$ else $\left\{\text{if } u < 1 - \frac{z_p}{z_i} \text{ then set } \xi_e^p = 0 \text{ else } \xi_e^p = 1\right\}$ where $u$ is a random between 0 and 1 |

(b)

| ID | prob. dist. | posterior-based proposal | |
|---|---|---|---|
| binomial $\xi_e \sim B(N, z)$ | $C_{\xi_e}^N z^{\xi_e}(1-z)^{N-\xi_e}$ where $C_n^N = \dfrac{N!}{n!(N-n)!}$ | $z_p > z_i$ $N_p = N_i$ | $\xi_e^p = \xi_e^i + X$ where $X \sim B\left(N_i - \xi_e^i, \frac{z_p - z_i}{1 - z_i}\right)$ |
| | | $z_p \leq z_i$ $N_p = N_i$ | $\xi_e^p \sim B\left(\xi_e^i, \frac{z_p}{z_i}\right)$ |
| | | $N_p > N_i$ $z_p = z_i$ | $\xi_e^p = \xi_e^i + X$ where $X \sim B(N_p - N_i, z_i)$ |
| | | $N_p \leq N_i$ $z_p = z_i$ | $\xi_e^p \sim HG(N_i, N_p, \xi_e^i)$ |

(*Continued.*)

**Table 1.** (*Continued.*)

| ID | prob. dist. | posterior-based proposal | |
|---|---|---|---|
| uniform $\xi_e \sim Uni(a,b)$ | $\left.\begin{array}{c}\frac{1}{b-a}\\0\end{array}\right\}\begin{array}{c}a\leq\xi_e\leq b\\\xi_e<a \text{ or } \xi_e>b\end{array}$ | | $\xi_e^p = a_p + \left(\frac{\xi_e^i-a}{b_i-a_i}\right)(b_p-a_p)$ |
| geometric $\xi_e \sim Geom(z)$ | $(1-z)^{\xi_e}z$ | $z_p > z_i$ | $X \sim Geom\left(\frac{z_p-z_i}{1-z_i}\right)$, if $X < \xi_e^i$ then $\xi_e^p = X$ else $\xi_e^p = \xi_e^i$ $u$ is a random number between 0 and 1 |
| | | $z_p \leq z_i$ | if $u < 1 - \frac{z_p}{z_i}$ then $\xi_e^p = \xi_e^i + 1 + X, X \sim Geom(z_p)$ else $\xi_e^p = \xi_e^i$ |
| negative binomial $\xi_e \sim NB(r,z)$ | $C_{\xi_e}^{\xi_e+r-1}z^{\xi_e}(1-z)^r$ where $C_n^N = \frac{N!}{n!(N-n)!}$ | $z_p > z_i$ $r_p = r_i$ | $\xi_e^p = \xi_e^i + q + X$ where $q \sim B(r_i, \frac{z_p-z_i}{1-z_i}), X \sim NB(q, z_p)$ |
| | | $z_p \leq z_i$ $r_p = r_i$ | $\xi_e^p \sim X$ where $X$ is drawn from p.m.f. $\sum_{q=1}^r (C_q^r C_{\xi_e^i-X-q}^{\xi_e^i-X-1}/C_{\xi_e^i}^{\xi_e^i+r-1})\left(1-\frac{z_p}{z_i}\right)^q\left(\frac{z_p}{z_i}\right)^X$ |
| | | $r_p > r_i$ $z_p = z_i$ | $\xi_e^p = \xi_e^i + X$ where $X \sim NB(r_p - r_i, z_i)$ |
| | | $r_p \leq r_i$ $z_p = z_i$ | $\xi_e^p \sim NHG(\xi_e^i + r_i - 1, \xi_e^i, r_p)$ |
| lognormal $\xi_e \sim lnorm(\mu, \sigma^2)$ | $\frac{1}{\xi_e\sqrt{2\pi\sigma^2}}e^{-\frac{(\log\xi_e-\mu)^2}{2\sigma^2}}$ | $\sigma_p > \sigma_i$ | $\log\xi_e^p \sim N(\mu_p + \alpha(\log\xi_e^i - \mu_i), \kappa(\sigma_p^2 - \sigma_i^2))$ where $\alpha^2 = \kappa + (1-\kappa)(\sigma_p^2/\sigma_i^2)$ |
| | | $\sigma_p \leq \sigma_i$ | $\log\xi_e^p \sim N\left(\mu_p + \alpha\frac{\sigma_p^2}{\sigma_i^2}(\log\xi_e^i - \mu_i), \kappa\frac{\sigma_p^2}{\sigma_i^2}(\sigma_i^2 - \sigma_p^2)\right)$ where $\alpha^2 = \kappa + (1-\kappa)(\sigma_i^2/\sigma_p^2)$ |

## 3.3. General approach

In general, the choice of ID will depend on the distribution $\pi(\xi \mid \theta^p, y)$, which itself is model dependent. If, for example, $\pi(\xi_e \mid \xi_{e' < e}, \theta, y)$ is better represented by a normal ID, the PBP sampling procedure is taken from the second line in table 1. In all, table 1 summarizes sampling schemes for twelve different ID functional forms, each corresponding to probability distributions commonly used in statistical models. These schemes are specifically designed to satisfy the following two conditions:

$$\text{Condition 1: } \frac{f_{\text{ID}}(\xi_e^p \mid \xi_{e' < e}^p, \theta^p, y)\, g(\xi_e^i)}{f_{\text{ID}}(\xi_e^i \mid \xi_{e' < e}^i, \theta^i, y)\, g(\xi_e^p)} = 1,$$

$$\text{Condition 2: } \xi_e^p \rightarrow \xi_e^i \text{ as } \theta^p \rightarrow \theta^i, \tag{3.4}$$

where $g(\xi_e^p)$ is the probability of sampling latent variable $\xi_e^p$ starting from state $i$, and $g(\xi_e^i)$ is the probability of sampling $\xi_e^i$ when proposing state $i$ from $p$. Condition 1 ensures that if $\xi_e^i$ is a random sample from $f_{\text{ID}}(\xi_e \mid \xi_{e' < e}^i, \theta^i, y)$ then $\xi_e^p$ will, by construction, be a random sample from $f_{\text{ID}}(\xi_e \mid \xi_{e' < e}^p, \theta^p, y)$ (albeit correlated with $\xi_e^i$). Condition 2 guarantees that proposals with small jumps in parameter space have an acceptance probability close to one.

Note, condition 1 can trivially be solved by sampling directly from the IDs

$$g(\xi_e^p) = f_{\text{ID}}(\xi_e^p \mid \xi_{e' < e}^p, \theta^p, y)$$

$$g(\xi_e^i) = f_{\text{ID}}(\xi_e^i \mid \xi_{e' < e}^i, \theta^i, y), \tag{3.5}$$

but this does not satisfy condition 2 and turns out to usually be inefficient.[7] Deriving sampling schemes which also satisfy condition 2 is a non-trivial task guided by intuition and trial and error. Extension of table 1 to encompass a more comprehensive list of possible sampling distributions will be the subject of future research. The validity of equation (3.4) for both Poisson and normal IDs is explicitly demonstrated in electronic supplementary material, appendix B.

## 3.4. Algorithm

We now describe the general algorithm used to implement PBPs:

**POSTERIOR-BASED PROPOSALS**

**Step 1: Generate $\theta^p$**—A proposed set of parameter values is drawn from a multivariate normal (MVN) distribution centred on the current set of parameters in the chain $\theta^i$

$$\theta^p \sim N(\theta^i, j^2 \Sigma^\theta), \tag{3.6}$$

where $\Sigma^\theta$ is a numerical approximation to the covariance matrix for $\pi(\theta \mid y)$ and $j$ is a tuneable jumping parameter (estimation of $\Sigma^\theta$ and optimization of $j$ are achieved during an initial 'adaptation' period, as explained in electronic supplementary material, appendix C). (Note, performing a joint update on all parameters instead of each individually helps to alleviate poor mixing due to strong parameter correlations in $\pi(\theta \mid y)$.)

**Step 2: Generate $\xi^p$**—We take each latent variable $\xi_e$ in turn (starting from $e = 1$ up to $e = E$) and calculate the characteristic quantities defining the IDs for the initial and proposed states (e.g. in the Poisson case this would be $\lambda_i$ and $\lambda_p$), as described in §4. $\xi_e^p$ is then sampled using specially designed proposals outlined in table 1 (note, this table contains separate lines referring to different potential ID functional forms). This sampling procedure is at the heart of PBPs and represents the key novelty of this approach.

**Step 3: Accept or reject joint proposal for $\theta^p$ and $\xi^p$**—With MH probability (see electronic supplementary material, appendix D)

$$P_{\text{MH}} = \min\left\{ 1, \frac{\pi(y \mid \xi^p, \theta^p)\, \pi(\xi^p \mid \theta^p)\, \pi(\theta^p)}{\pi(y \mid \xi^i, \theta^i)\, \pi(\xi^i \mid \theta^i)\, \pi(\theta^i)} \prod_{e=1}^{E} \frac{f_{\text{ID}}(\xi_e^i \mid \xi_{e' < e}^i, \theta^i, y)}{f_{\text{ID}}(\xi_e^p \mid \xi_{e' < e}^p, \theta^p, y)} \right\}. \tag{3.7}$$

---

[7]This approach is akin to standard importance sampling and typically leads to very low acceptance rates for high dimensional models (as observed in examples 5.2 and 5.3 later).

The algorithm above performs a random walk through the parameter space defined by the posterior, but because the dimensionality of $\theta$ is typically much less than $\xi$, it is expected to mix at a much faster rate[8] than standard MCMC, which performs a random walk in both $\theta$ and $\xi$. Further insights into the PBP procedure are given in electronic supplementary material, appendix E.

PBPs, by design, result in $\xi^p$ exhibiting correlation with $\xi^i$ (indeed, if $\theta^p = \theta^i$ then $\xi^p$ is exactly the same as $\xi^i$, as required by equation (3.4)). The PBP MCMC algorithm used in this paper mitigates against these correlations by performing PBPs interspersed with standard updates for the latent variables[9] every $U$ steps. Electronic supplementary material, appendix H shows that mixing is not sensitive to the exact value of $U$, and is approximately optimized when $U = 4$ (as subsequently used).[10]

# 4. Generating importance distributions

For each latent variable $\xi_e$, the distribution we wish to approximate is $\pi(\xi_e | \xi_{e' < e}, \theta, y)$, i.e. the posterior probability distribution for $\xi_e$ given $\xi_{e' < e}$, parameters $\theta$, and data $y$. This distribution can be expressed as

$$\pi(\xi_e | \xi_{e' < e}, \theta, y) = \int \int \pi(\xi_{d \geq e} | \xi_{e' < e}, \theta, y) \mathrm{d}\xi_E \ldots \mathrm{d}\xi_{e+1}, \tag{4.1}$$

where $\pi(\xi_{d \geq e} | \xi_{e' < e}, \theta, y)$ is the joint posterior distribution for latent variables with index $e$ and above (conditional on everything else). The integrals in equation (4.1) successively marginalize over the unknown latent variables, starting with the last $\xi_E$ all the way back to $\xi_{e+1}$. Using Bayes' theorem and equation (1.1), the integrand in equation (4.1) can be expressed as

$$\pi(\xi_{d \geq e} | \xi_{e' < e}, \theta, y) \propto \pi(y | \xi, \theta) \pi(\xi_{d \geq e} | \xi_{e' < e}, \theta)$$
$$\propto \pi(\xi_e | \xi_{e' < e}, \theta) \pi(y | \xi, \theta) \prod_{d=e+1}^{E} \pi(\xi_d | \xi_{e' < d}, \theta). \tag{4.2}$$

For now, making the simplification that one observation is made per latent variable, i.e.

$$\pi(y | \xi, \theta) = \prod_{e=1}^{E} \pi(y_e | \xi_e, \theta), \tag{4.3}$$

and substituting equation (4.2) into equation (4.1) gives

$$\pi(\xi_e | \xi_{e' < e}, \theta, y) \propto \pi(\xi_e | \xi_{e' < e}, \theta) \pi(y_e | \xi_e, \theta)$$
$$\times \int \int \prod_{d=e+1}^{E} \pi(\xi_d | \xi_{e' < d}, \theta) \pi(y_d | \xi_d. \theta) \mathrm{d}\xi_E \ldots \mathrm{d}\xi_{e+1}. \tag{4.4}$$

Analytically performing these integrals is usually not possible. However, different levels of approximation can be made depending on the point at which the expression on the right-hand side of equation (4.4) is truncated. This leads to a family of importance distributions with increasing accuracy (as illustrated in figure 4).

## 4.1. ID$_0$

By taking just the first term on the right-hand side of equation (4.4), the ID is simply set to the distribution from the model itself[11]

$$f_{\mathrm{ID}_0}(\xi_e | \xi_{e' < e}, \theta) = \pi(\xi_e | \xi_{e' < e}, \theta). \tag{4.5}$$

Note, this can only be done provided the model distribution $\pi(\xi_e | \xi_{e' < e}, \theta)$ has a functional form belonging to one of the possibilities in table 1 (or alternatively a new PBP sampling scheme based on the model distribution is created which satisfies the conditions in equation (3.4)). If this cannot be achieved, the functional form for ID$_0$ is chosen to match $\pi(\xi_e | \xi_{e' < e}, \theta)$ as closely as possible.

The calculation of ID$_0$, as illustrated in figure 4b, involves only those latent variables on which $\xi_e$ is conditionally dependent. PBPs using ID$_0$ are equivalent to model-based proposals (MBPs) [9], and their

---

[8]Subject to sufficiently large jumping size $j$ in equation (3.6) being possible.

[9]Which passes through each latent variable and performs a Gibbs or random walk MH update.

[10]In practice, the optimum $U$ will depend on the relative CPU time needed for PBP and standard updates. If standard updates are much slower it makes sense for $U$ to be higher, but typically they are of a similar speed.

[11]Note, the proportionality sign in equation (4.4) becomes an equality sign because $\pi(\xi_e | \xi_{e' < e}, \theta)$ is normalized.

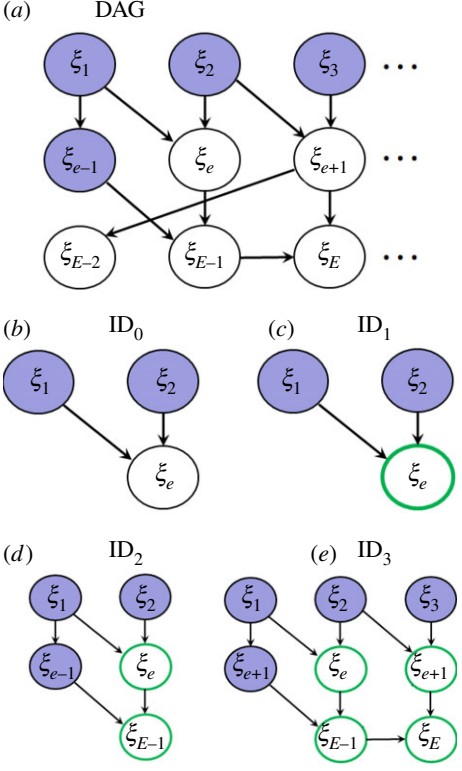

**Figure 4.** Importance distributions (ID). (a) The model (circles containing parameters and observations have been omitted for clarity). The posterior distribution for $\xi_e$ assumes that $\xi_{e'<e}$ are known (indicated by the blue shading). (b–e) Successive approximations for the ID (described in the text). The bold, green circles indicate that observations on these latent variables have been used in calculating the corresponding ID.

MH acceptance probability from equation (3.7) simplifies to

$$P_{MH} = \min\left\{1, \frac{\pi(y|\xi^p,\theta^p)\pi(\theta^p)}{\pi(y|\xi^i,\theta^i)\pi(\theta^i)}\right\}. \tag{4.6}$$

One of the desirable features of MBPs is that they require no hand-tuning. There is a one-to-one correspondence between the model distributions and the proposals, as outlined in table 1, and so they can be implemented in an automated manner. However, in cases in which data substantially restricts model parameters and latent variables, higher-order importance distributions become necessary.

## 4.2. ID$_1$

The ID accuracy is improved by using the first two terms on the right-hand side of equation (4.4)

$$f_{ID_1}(\xi_e|\xi_{e'<e},\theta,y) = c\,\pi(\xi_e|\xi_{e'<e},\theta)\,\pi(y_e|\xi_e,\theta), \tag{4.7}$$

where $c$ is a normalizing factor. Calculation of ID$_1$, as illustrated in figure 4c, includes not only those latent variables on which $\xi_e$ is dependent, but also the observation $y_e$ on $\xi_e$ itself (as indicated by the green circle).[12]

## 4.3. ID$_2$

This additionally includes observations on those latent variables which depend on $\xi_e$, e.g. $\xi_{E-1}$ in figure 4d. Equation (4.4) now gives the improved approximation

$$f_{ID_2}(\xi_e|\xi_{e'<e},\theta,y) = c\,\pi(\xi_e|\xi_{e'<e},\theta)\,\pi(y_e|\xi_e,\theta)\int\pi(\xi_{E-1}|\xi_{e'<E-1},\theta)\,\pi(y_{E-1}|\xi_{E-1},\theta)\mathrm{d}\xi_{E-1}. \tag{4.8}$$

---

[12]This relies on the product of model and observation probability distributions being contained within table 1. If this is not the case then some level of approximation is necessary.

Higher-order approximations (i.e. $ID_{n>2}$ which take into account successively more of the model and data[13]) usually prove to be too computationally expensive to be efficient.

## 4.4. Choosing ID

Choosing which level of ID to optimize PBPs involves a trade-off between the computational cost of generating IDs with the size of posterior jumps (and hence improvement in mixing) they allow. Unfortunately determining *a priori* which option is best is challenging. Indeed, in the results section below we find examples for which $ID_0$, $ID_1$ and $ID_2$ each represent optimum solutions for different problems. From the point of view of the user, the pragmatic approach to take is to first try $ID_0$ (which is the easiest to implement) and if that doesn't help mixing then try $ID_1$ and so on and so forth. Identification of optimal IDs will be the subject of active future research.

The classification scheme presented above is based on models which have one observation per latent variable. For models in which this is not the case, generation of IDs relies on approximating the following expression:

$$\pi(\xi_e|\xi_{e'<e},\theta,y) \propto \pi(\xi_e|\xi_{e'<e},\theta) \int \int \pi(y|\xi,\theta) \prod_{d=e+1}^{E} \pi(\xi_d|\xi_{e'<d},\theta) d\xi_E \ldots d\xi_{e+1}. \tag{4.9}$$

This may or may not be computationally challenging, depending on the scenario considered. However, provided the model itself makes uses of the distributions in table 1, using $ID_0$ (i.e. MBPs) is always possible.

# 5. Empirical evaluation

We now investigate the relative computational performance of PBPs compared to other approaches (where appropriate):

*Standard MCMC*—Here an 'update' is performed by sequentially making changes to each model parameter and latent variable in turn. Where possible, Gibbs sampling [7] is used, otherwise random walk MH is implemented (note, computational efficiency is optimized by calculating only those parts of the likelihood and observation probability which actually change given a particular proposal).

*Non-centred parametrization* (NCP)—Standard approaches can also be applied to so-called non-centred parametrizations (NCPs). Here, inference is performed on $(\theta,\xi')$, where $\xi'$ are distributed independently of $\theta$ and the actual latent variables are related through deterministic relationship $\xi = h(\xi',\theta,y)$ [26]. To give a simple example, suppose each latent variable is distributed normally $\xi_e \sim N(\mu,\sigma^2)$ with mean $\mu$ and variance $\sigma^2$ being model parameters $\theta$. This can be reparametrized by setting $\xi'_e \sim N(0,1)$, with the functional dependency $h$ being given by $\xi_e = \mu + \sigma\xi'_e$. Note, NCPs are not always possible because parameters cannot always be separated from distributions, (e.g. this cannot be done for the Bernoulli, Poisson or Gamma distributions), hence NCPs cannot be used in examples 5.1 and 5.3 below. More complicated schemes which make use of partial CP/NCP proposals and interweaving different parametrizations [4,16,17,27] are not considered here.

*Hamiltonian MCMC* (HMCMC)—This generates samples by integrating a trajectory from the current parameter and latent variable state to a proposed state (typically via many intermediary steps) [3,12]. Such a process accounts for gradients in the log of the posterior probability, allowing large distances in parameter and latent variable space to be traversed. Optimization balances making the initial and proposed states as uncorrelated as possible (to improve mixing), while reducing the computational burden of excessive steps and allowing for a sufficiently good acceptance probability. The Metropolis-adjusted Langevin algorithm (MALA) [28] is a special case of HMCMC in which only a single step is taken. HMCMC is limited to only those models with continuous model parameters and latent variables, and consequently cannot be applied to examples 5.1 and 5.4 below.

*Particle MCMC* (PMCMC)—This approach generates unbiased approximations $\widehat{\pi}(y|\theta)$ by means of a sequential filtering process [2]. In its simplest implementation this involves running multiple simulations of the model (i.e. sampling from $ID_0$ in §4) which are periodically filtered based on the observations; however, more efficient schemes can make use of higher-order importance distributions $ID_1$, $ID_2$, etc.

---

[13]$ID_n$ is the approximation to equation (4.4) in which those latent variables $\xi_d$ that are $n-1$ or fewer arrows away from $\xi_e$ (along with any latent variables on which they depend) are included in the integral.

Jumping in parameter space (e.g. using equation (3.6)) is achieved through a MH algorithm that makes use of these unbiased estimates.

Further details, along with optimization procedures for each of the different methods, are described in electronic supplementary material, appendices I–K.

Additionally, it should be pointed out that another technique to improve mixing is to simply integrate out problem parameters directly. Such an approach, however, is not considered in this paper for two reasons: firstly, it lacks generality, because it only applies to models in which these integrals can actually be performed, and secondly, it restricts the possible priors that can be applied to a given model (most prior choices make such integration impossible).

Four contrasting model types are investigated. In each case, we examine the efficiency of the various algorithms for parameter inference from data simulated from the model in question. Results shown are based on $10^6$ MCMC updates preceded by $10^4$ discarded samples from a burn-in/adaptation period (see electronic supplementary material, appendix C). One way to measure MCMC efficiency is to calculate the 'effective sample size' [1] (see electronic supplementary material, appendix L), and here we calculate the computational time for any given algorithm to generate 100 effective posterior samples[14] of a given parameter.[15] Unless otherwise stated, uninformative flat priors are assumed.

## 5.1. Inferring disease prevalence and diagnostic test performance

Suppose we aim to estimate the disease prevalence (fraction of infected) $p_D$ in a population of $P$ individuals using cross-sectional diagnostic test results. Such diagnostic tests are typically imperfect, and characterized by a sensitivity $Se$ (the probability of a positive test result given an infected individual) and specificity $Sp$ (the probability of a negative test result given uninfected). Suppose $Se$ and $Sp$ are unknown. In the absence of a gold standard defining which individuals are truly infected, inference is only possible when two or more independent test results are recorded per individual (due to confounding). Here, we assume that results are available from a single diagnostic test performed on each individual at two times, labelled $t = \{1,2\}$. The model is shown in figure 5$a$ and described in detail along with the development of PBP proposals in electronic supplementary material, appendix M. We note that this model could be embedded in more complex models, e.g. fitted to data from capture-mark-recapture programmes [29].

### 5.1.1. Speed comparison

Simulated data was created using $p_D = 0.5$, $Se_1 = Se_2 = 0.6$ and $Sp_1 = Sp_2 = 0.9$ for $p = 1000$ individuals. Inference was then performed using MBP/PBP MCMC approaches as well as standard Gibbs sampling. Figure 5$b$ shows how posterior samples for $p_D$ vary as each of the three algorithms progress. By binning these samples, the marginal posterior distributions for $p_D$ can be generated, as shown in figure 5$c$ (which also shows results for the other model parameters). These distributions contain the known values used to generate the data (denoted by vertical black lines), indicating successful inference.

It is important to note that, in the limit of infinite MCMC sample number, all algorithms generate exactly the same set of marginal distributions (i.e. those shown in figure 5$c$). However, the speed with which they converge *does* vary. For example, figure 5$d$ shows how the computational time (i.e. the CPU time to generate 100 effective samples) to infer $p_D$ increases with population size. Here we find that MBPs (ID$_0$) actually performs worse than the standard Gibbs approach; however, when observations are incorporated (ID$_1$) the resultant PBP algorithm is around three times faster than Gibbs sampling (at least when the number of individuals is large). Although mixing is greatly improved, as is evident in figure 5$b$, this is offset by the computational overhead associated with each PBP.

PMCMCs which use ID$_0$ (i.e. the usual approach of simulating from the model) are found to perform very poorly (this is because a very large number of particles are required for a reasonable acceptance rate). However, when ID$_1$ is used, PMCMC actually becomes the fastest approach. In this particularly simple example, ID$_1$ exactly represents the posterior and so PMCMC only requires a single particle to run. This is atypical, and usually particle methods fail when large numbers of observations are made (as demonstrated later). Because this model contains discrete latent variables (i.e. the underlying

---

[14]Note, all MCMC chains were run long enough to be well mixed, with ESS typically exceeding 1000 and at least greater than 500.

[15]Simulation and inference were averaged over 20 separate runs to help remove data-dependent noise.

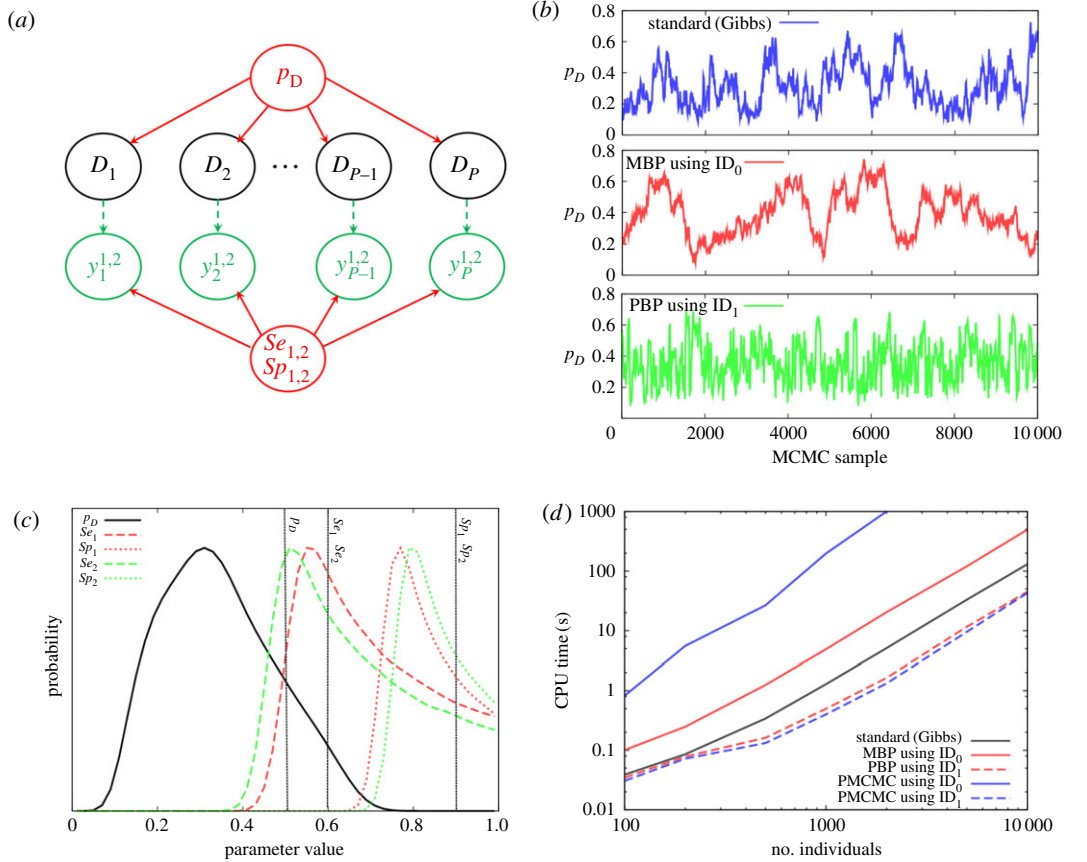

**Figure 5.** Disease diagnostic test model. (*a*) The DAG with probability of disease $p_D$, true disease status $D_e$ (1/0 denoting infected/uninfected), and observed disease status as measured from two diagnostic tests $y_e^{1,2}$ (which have sensitivities $Se_{1,2}$ and specificities $Sp_{1,2}$). (*b*) Trace plots for disease probability $p_D$ as a function of MCMC sample for three different algorithms. (*c*) The posterior distributions for the model parameters (the vertical lines represent the true values). (*d*) Shows how the CPU time needed to generate 100 effective samples for $p_D$ changes as a function of the number of individuals in the population for various different approaches.

disease status of individuals $D_e$ is 0 or 1), HMCMC approaches are not possible, and the Bernoulli distribution does not allow for NCP.

This example demonstrates a possible modest improvement in computational speed when using PBPs compared to a standard Gibbs approach. The next example, however, shows that much larger potential gains can be made.

## 5.2. Stochastic volatility model

Stochastic volatility (SV) models are used to capture time-varying volatility on financial markets, and are essential tools in risk management, asset pricing and asset allocation [30]. In economics, a 'logarithmic rate of return' can be defined by $y_e = \log(V_{e+1}/V_e)$, where $V_e$ is the price of an asset (e.g. a share) on day $e$. Consequently, when $y_e$ is positive it means that on day $e+1$ the asset goes up in price, but when it is negative it goes down. One way to capture time variation in $y_e$ is through the so-called SV$t$ model [31]

$$
\begin{aligned}
y_e &= e^{h_e/2} u_e, \\
h_e &= \mu + \phi(h_{e-1} - \mu) + \eta_e,
\end{aligned}
\tag{5.1}
$$

where $u_e$ are independent and identically distributed (i.i.d.) with a Student's $t$-distribution (characterized by parameter $v$) and $\eta_e$ are i.i.d. normal (with zero mean and variance $\sigma^2$). Note, if $h_e$ is fixed (i.e. $\sigma = 0$), the logarithmic rate of return $y_e$ would simply be sampled asymptotically from a distribution with fixed variance, or 'volatility'. The introduction of time variation in the variable $h_e$, whose temporal correlations are measured by $0 < \phi < 1$, means that $y_e$ experiences SV, i.e. periods when there are large variations in asset price, and other periods when there is not much variation.

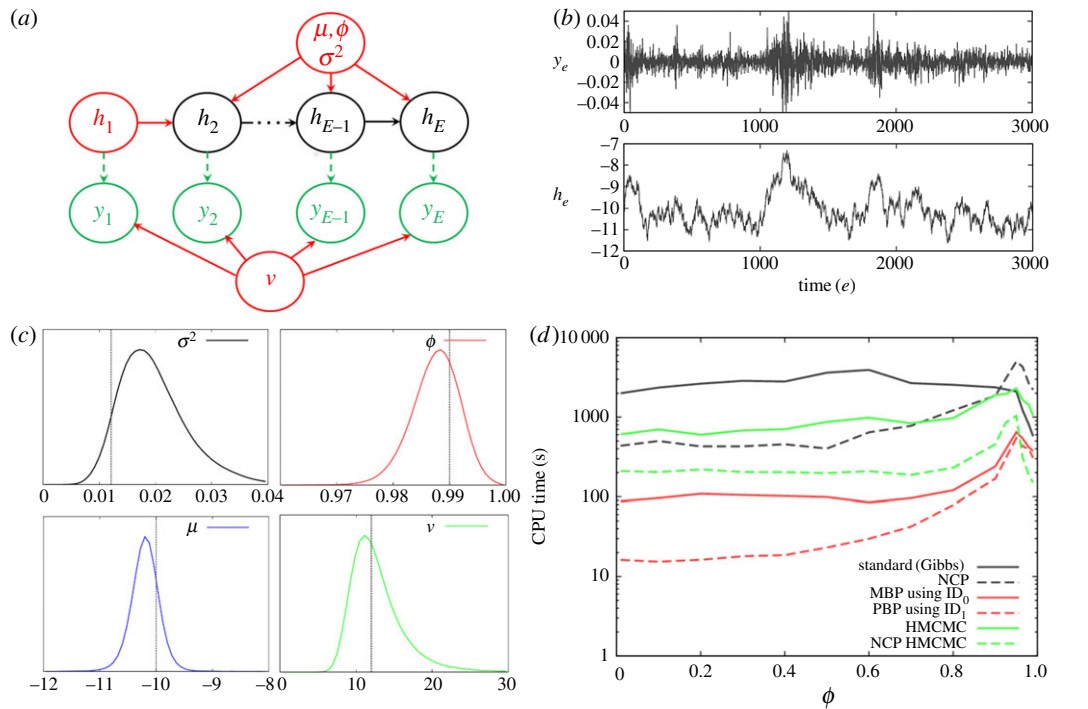

**Figure 6.** Stochastic volatility model. (*a*) The DAG (see §5.2 for details). (*b*) Simulated data. (*c*) Posterior distributions for model parameters (the vertical lines represent the true values) based on the simulated data. (*d*) Shows how the CPU time needed to generate 100 effective samples of $\sigma^2$ varies as the correlation parameter $\phi$ used to simulate the data changes.

The SV$t$ model is represented in figure 6$a$. Simulated data were created using $\mu = -10$, $\phi = 0.99$, $v = 12$, $\sigma^2 = 0.0121$ (which are parameter values based on estimates made from the S&P 500 index [30]). Figure 6$b$ shows the time variation in $y_e$ and $h_e$ (observe how changes in $h_e$ correspond to changes in the volatility of $y_e$) over $E = 3000$ days. A detailed development of PBP proposals for this model is given in electronic supplementary material, appendix N.

### 5.2.1. Speed comparison

Bayesian inference from the simulated data in figure 6$c$ identifies marginal posterior distributions which contain the true parameter values, as indicated by black vertical lines. For each algorithm, figure 6$d$ shows how the computational time to infer $\sigma^2$ varies with the correlation parameter $\phi$ used to generate the data (with all other parameters fixed as above).

The standard algorithm is at its fastest when $\phi \sim 1$ but slows down considerably as $\phi$ is reduced. Some speed-up is observed when NCP is used (where $h'_e = (h_e - \mu)/\sigma$), however, the MBP and PBP methods are found to be much faster (note PBP using $ID_2$ was not found to be any faster than with $ID_1$, and so is not shown). HMCMC was found to be relatively slow using the standard parametrization, but markedly increased in speed when using NCP. PMCMC methods (using either $ID_0$ or $ID_1$) were found to be extremely slow (because they required a huge number of particles), and lie above the top of this graph.

For real financial markets, $\phi$ is within the range of 0.95–0.99 [30], reflecting a high degree of persistence in volatility. This corresponds to PBPs running between two and four times faster than the standard approach, and comparable in speed with HMCMC (sometimes faster and sometimes slower).[16] However, the left-hand side of figure 6$d$ clearly demonstrates the existence of regimes for which PBPs are faster by a factor exceeding 10 than all other methods tried.

## 5.3. Mixed model

MMs [32] explain observations in terms of both 'fixed effects' (e.g. individual attributes such as gender or disease status) and 'random effects' (which account for random uncontrollable factors within a study, e.g.

---

[16]The reason PBPs become slower as $\phi \to 1$ is that (due to correlations introduced by $\phi$) $\pi(\xi_e \mid \xi_{e' < e}, \theta, y)$ is expected to depend on observations up to around $1/(1 - \phi)$ days ahead. In contrast, $ID_n$ only includes observations up to $n - 1$ days ahead, which is typically a much shorter interval.

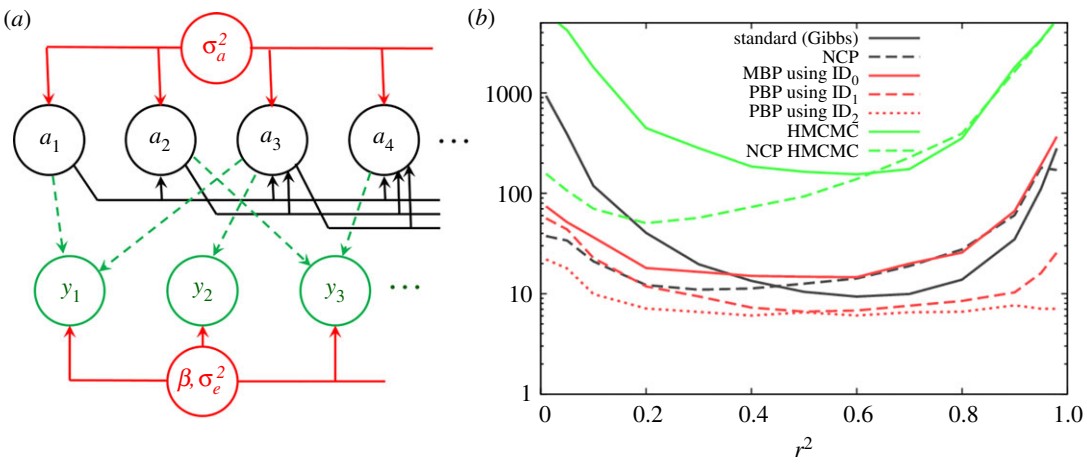

**Figure 7.** Mixed model. This looks at a mixed model applied to a quantitative genetics application. (a) The DAG where $a$ are multivariate normally distributed with covariance matrix $\sigma_a^2\mathbf{A}$ (**A** is the relationship matrix), vector $\beta$ are fixed effects and $y$ are trait observations with residual covariance matrix $\sigma_a^2\mathbf{I}$ (see §5.3 for further details). (b) Shows how the CPU time needed to generate 100 effective samples of $r^2 = \sigma_a^2/(\sigma_a^2 + \sigma_\varepsilon^2)$ (which characterizes the genetic heritability) varies as the $r^2$ used to simulate the data changes.

variation in student grades as a result of variation in the quality of schools). MMs are useful in a wide variety of applications in the physical, biological and social sciences [33–36]. They assume that a vector of $N$ measurements $y$ can be decomposed into three contributions:

$$y = X\beta + Za + \varepsilon, \tag{5.2}$$

where $X$ and $Z$ are design matrices that define model structure, $\beta$ is a vector of $F$ fixed effects, $a$ is a vector of $E$ random effects and $\varepsilon$ are residuals. Random effects and residuals are assumed to be MVN with zero mean and covariance matrices $G$ and $R$, respectively. For simplicity, we assume that $G = \sigma_a^2 A$, where $A$ is a known symmetric matrix, and $R = \sigma_\varepsilon^2 I$, where $I$ is the identity matrix (such that residuals are uncorrelated between measurements). The quantity $r^2 = \sigma_a^2/(\sigma_a^2 + \sigma_\varepsilon^2)$ measures the relative contribution of random effects to residuals.

### 5.3.1. Speed comparison

An important application of MMs is in the field of quantitative genetics, which aims to understand the genetic basis of traits of interest [37]. As an illustration take $y$ to represent measurements of height within a population. These measurements are correlated, e.g. if an individual has tall parents, they are more likely to be tall. This correlation results from genetic inheritance. The relatedness of individuals within the population is captured by the so-called relationship matrix $A$.

The DAG for the model is shown in figure 7a and simulated data were generated assuming a population randomly mated over four generations with $N = E = 4 \times 10^3$ individuals and $F = 2$ fixed effects (see electronic supplementary material, appendix O for further details). Figure 7b shows how the computational time to infer $r^2$ varies with the $r^2$ value used to generate the data. Disregarding fixed effects, $r^2 = 0$ implies no genetic inheritance and $r^2 = 1$ corresponds to a trait dominated by genetic inheritance. In these limits, Gibbs sampling slows down significantly due to strong parameter-latent variable correlations in the posterior, leading to poor mixing. Using MBPs with $ID_0$ shows an improvement over Gibbs for low $r^2$; however, for high $r^2$, its performance proves to be poor. Using $ID_1$ and $ID_2$ leads to further improvements in computational speed, resulting in PBPs becoming consistently faster than the standard Gibbs approach (e.g. $ID_2$ is a factor ~50 times faster in the limit $r^2 \rightarrow 0$). The move to $ID_3$ shows little improvement and for $ID_4$, $ID_5$ etc. PBPs become progressively slower despite mixing better. Consequently, $ID_2$, which uses measurements taken on individuals as well as close relatives, represents an optimum choice in this particular example.

Results for HMCMC in figure 7b are very slow. The reason lies in the fact that the trajectories themselves are found to behave diffusively (e.g. if a trace plot of $\sigma_a^2$ is made, its path exhibits familiar random walk behaviour, rather than the relatively smooth progress from one side of the posterior to the other that would be hoped for). NCP HMCMC (which sets $a' = a/\sigma_a$) led to a marked improvement, but still it remained considerably slower than the other methods.

One of the striking features of figure 7$b$ is how well the NCP standard approach works (only around two times slower than the best PBPs for lower $r^2$). The reason is that for this particular model NCP and MBPs work in much the same way: under NCP, proposals in $\sigma_a$ lead to a simultaneous expansion or contraction of the random effects (through reparametrization $\boldsymbol{a} = \sigma_a \boldsymbol{a}'$), and this is also what happens in MBPs when the value of $\kappa$ in table 1 is set to zero (note, the two curves for these methods lie very close to each other in figure 7$b$, with NCP slightly faster due to the fact that fewer computations are required per update).

PBP and NCP approaches have also been applied to MMs and generalized MMs (binary disease data) with diagonal, sparse and dense $A$ (results not shown) and overall PBPs were not found to perform substantially faster than NCP, and in some cases were slower. Consequently, for these types of model, standard approaches using NCP may prove to be the best method to use, particularly given the relative ease with which they can be implemented (table 2).

## 5.4. Logistic population model

We here consider a simple illustrative example taken from ecology. Imagine time is discretized in intervals of size $\tau$ and suppose we are following the population size of an animal species which has been released into the wild. We know that births and deaths will occur, and that the population size will increase, but that increase will be curtailed by the limited resources within the area. This can be modelled in the following way

$$
\begin{aligned}
b_t &\sim \text{Pois}\big(\tau r_b P_t (1 - P_t/K)\big), \\
d_t &\sim \text{Pois}(\tau \mu P_t), \\
P_{t+1} &= P_t + b_t - d_t,
\end{aligned}
\tag{5.3}
$$

where $b_t$, $d_t$ and $P_t$ are the number of births, deaths and population size in time interval $t$, $\text{Pois}(\lambda)$ generates Poisson distributed integer samples with mean $\lambda$, $r_b$ is the birth rate, $\mu$ is the mortality rate and $K$ is the carrying capacity (which determines the maximum size of the population). Equation (5.3) can be considered as a Tau-leaping approximation to the underlying continuous time process under study [38].

A DAG for this model is shown in figure 8$a$ and a simulation is shown in figure 8$b$. Now suppose that to keep track of this wildlife population, traps are set at certain points in time and the number of trapped individuals is recorded (shown by the red crosses).[17] Note, animals are caught with capture probability $p$, and so these result are much less that the actual population sizes and also contain additional stochastic noise.

The data from figure 8$b$ alone is insufficient to estimate all four model parameters, so here semi-informative priors are placed on the mortality rate $\mu$ and capture probability $p$ (in reality captured animals are marked and then when re-captured this provides direct evidence for these quantities).[18] Figure 8$c$ shows the results of inference, with $\mu$ and $p$ largely following their prior distributions and reasonably good estimates being obtained for birth rate $r_b$ and carrying capacity $K$.

### 5.4.1. Speed comparison

The CPU time to estimate 100 effective samples of $r_b$ is shown in figure 8$d$ as a function of the number of measurements made during the time interval. On the right-hand side is the extreme case in which measurements are made at every single time point. Here the observations tightly restrict the potential values for the latent variables, and the standard approach is actually found to perform best. On the other hand, as fewer and fewer observations are made, both the MBP and PMCMC methods become more and more efficient. Note, however, that MBPs are consistently around five times faster than PMCMC. Despite mixing faster, PMCMCs take much longer per update because of the large number of particles (simulations of the process) needed.

Unfortunately here PBPs are not found to be effective because identification of the importance distribution at time $t$ is informed by the next measurement, which is potentially many time steps into the future. Further work is need to develop effective IDs in these types of situation, which go beyond the simple classification scheme from §4.

---

[17]Trapped animals are then released back into the wild.

[18]Specifically a gamma-distributed prior on $\mu$ with mean 0.3 and variance 0.0144 and a beta distributed prior on $p$ with mean 0.5 and variance 0.0025.

**Table 2.** This table gives a brief description of the methods along with various pros and cons. Here, 'data dominant' refers to situations in which the number of observations is similar (or exceeding) the number of latent variables and 'model dominant' relates to the converse case in which few observations are made on a model containing many latent variables. MH stands for Metropolis–Hastings, and CP and NCP stand for centred and non-centred parametrizations.

| method | description | performance | limitation | optimization/implementation |
|---|---|---|---|---|
| MBP | Generate $\theta^p$ from $\theta^i$ and modify $\xi^i$ to generate $\xi^p$ (based on the model). | Found to exhibit faster mixing than standard approach in a large number of scenarios, particularly those which are 'model dominant'. | Requires that the distributions used in the model are found in table 1 (or can be derived using the conditions in equation (3.4)). | Easy—the only free parameter is $U$ (which governs the relative rate of MBP to standard updates), and results are found to be relatively insensitive to its value. |
| PBP | Generate $\theta^p$ from $\theta^i$ and modify $\xi^i$ to generate $\xi^p$ (based on importance distributions that account for the data). | Found to be the fastest approach in many cases. Provides additional computational performance compared to MBPs, especially in the 'data dominant' regime. | Requires that the importance distributions used are found in table 1 (or can be derived using the conditions in equation (3.4)). | Medium–hard—the most challenging aspect of PBPs is obtaining good importance distributions that approximate the posterior. In the case of $ID_1$ this is often relatively straightforward, but higher order approximations can be more difficult, if not impossible, to identify. |
| HMCMC | Randomly sample momentum vector at $(\theta^i/\xi^i)$ and integrate Hamilton's equations to generate $(\theta^p, \xi^p)$. | Can work very well under certain circumstances (e.g. it was the fastest approach in figure 6d); however, can be slow if integrated trajectories behave diffusively (which results in a very large number of steps being necessary to pass from one side of the posterior to the other). | Only works with continuous parameters and latent variables and in situations in which the gradients in the log-likelihood can be calculated. | Hard—optimizing the step size is relatively easy, as it can be selected to achieve a certain acceptance rate. Optimizing the number of steps for each update, however, is difficult and efficiency is found to critically depend on this value. Automated methods such as NUTs have been developed, but these are not easy to implement. Also choice of CP/NCP was found to substantially affect performance. |

(*Continued.*)

**Table 2.** (*Continued.*)

| method | description | performance | limitation | optimization/implementation |
|---|---|---|---|---|
| PMCMC | Generate $\theta^p$ from $\theta^i$ and use particles to generate unbiased estimate $\widehat{\pi}(y\|\theta)$. This is then used in a MH update. | These were found to work reasonably well in the 'model dominant' regime. In the idealized case when latent variable can be exactly sampled (requiring only a single particle) they are fastest, but in the other examples investigated MBPs/PBPs were found to outperform. | Observations on the system need to be sequentially ordered (such as in time-series data) to allow for particle filtering to work. | Easy–hard—simulation from the model and subsequent particle filtering usually straightforward. However, things are more complicated when importance distribution are required (to avoid an unreasonably large number of particles), similar to the challenges faced by PBPs. |
| Stand. / NCP | Make local changes to $\theta$ and $\xi$. Gibbs sampling is performed where possible, otherwise random-walk MH updates. | Standard approaches tend to work well in the 'data dominant' regime (here parameter and latent variable values are well established and correlations between them are less important. Using NCP, where possible, was often found to substantially increase speed. | none | Easy–medium—when performing local updates, care is needed to only calculate those parts of the likelihood and observation model which change. Often CP and NCP work in different regimes, hence overall optimization would require performing combinations of the two. |

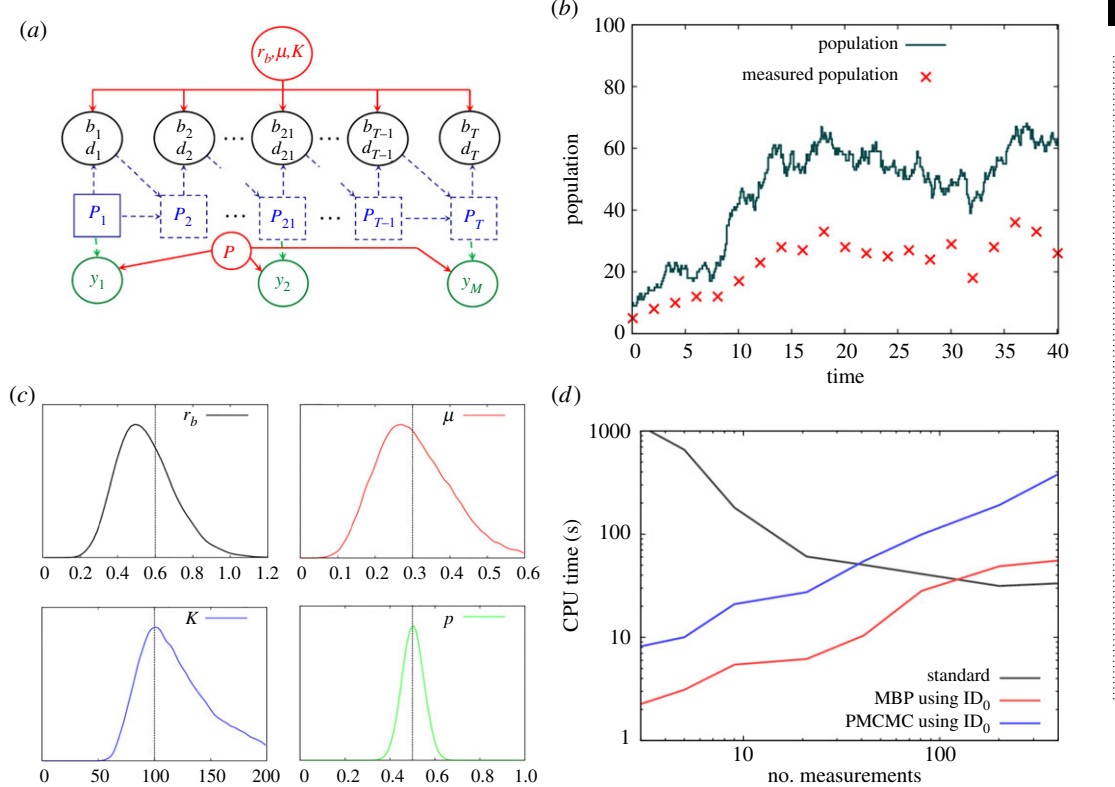

**Figure 8.** Logistic population model. (*a*) The DAG, where $r_b$ is the birth rate, $\mu$ is the mortality rate, $K$ is the carrying capacity, $b_t$ and $d_t$ are the number of births and deaths at discrete time interval $t$ (which runs from 1 to $T$), $P_t$ is the population size, and $y_m$ (where $m$ runs from 1 to $M$) are the numbers of trapped individuals at various measurement points ($p$ is the trapping probability). (*b*) Simulated data, where $T = 401$ and $\tau = 0.1$ is the time-step size, $r_b = 0.6$, $\mu = 0.3$, $K = 100$, $p = 0.5$ are the model parameters and $M = 21$ is the number of measurements. (*c*) The posterior distributions for the model parameters (the vertical lines represent the true values) based on this simulated data. (*d*) Shows how the CPU time needed to generate 100 effective samples of $r_b$ varies as the number of equally spaced measurements $M$ changes.

Note, HMCMC was not possible because of the discrete latent variables in the model and NCP methods could not be used because model parameters cannot be separated from Poisson-distributed latent variables.

## 6. Summary

This paper introduced PBPs and demonstrated that they speed up MCMC inference in many cases where existing approaches perform poorly. PBPs are applicable to the majority of statistical models (namely, those whose conditional dependence structure is expressible in terms of a directed acyclic graph). Performance is enhanced by improving mixing of MCMC (i.e. increasing the rate at which they generate uncorrelated samples) by jointly proposing changes to model parameters $\theta$ and latent variables $\xi$ (see §3 for details). PBPs are a family of proposal schemes built by generating importance distributions ($ID_0$, $ID_1$, etc.) that systematically account for dependence structure in the Bayesian posterior with increasing accuracy. The zeroth-order approximation $ID_0$ ignores the data and thus corresponds to MBPs [9]. The optimal level of approximation depends on problem-specific trade-offs between improved mixing resulting from increased acceptance rates and the computational cost of generating suitably accurate IDs.

The relative computational speed of PBPs compared to 'standard' Gibbs/random walk MH techniques (using centred and non-centred parametrizations) as well as HMCMC and PMCMC approaches was investigated for various benchmark models used in applications ranging from finance to ecology to statistical genetics. While different methods worked better or worse in different scenarios, PBPs were found to be either near to the fastest or significantly faster than the next best approach (by up to a factor of 10). Table 2 summarizes the relative strengths and weaknesses of the various approaches.

# 7. Discussion

We now discuss PBPs in relation to PMCMC and HMCMC under two regimes: 'model dominant' and 'data dominant'. Here, we define 'model dominant' to relate to problems in which the shape of the posterior is largely represented by the latent process likelihood, with the observation model providing a small perturbation on top of this. A good example of this would be a complex model applied to relatively few actual measurements (e.g. the left-hand side of figure 8*d*). The flip side of this is the 'data dominant' regime, in which the data exceeds or is comparable to the model complexity[19] (e.g. the right-hand side of figure 8*d*).

As stated previously, PMCMC only works on problems in which data can be incorporated in a sequential manner. Even then, however, PMCMC can become slow in the data dominant regime due to requiring a very large number of particles to give a reasonable acceptance rate.[20] This was explicitly demonstrated in examples 5.2 and 5.3 above, where PMCMC was found to be vastly slower than the other approaches. However, a key advantage of PMCMC approaches is that they are usually relatively easy to implement and typically allow for efficient parallelization.

HMCMC relies on calculating gradients in the log-likelihood, and hence is not applicable to models with discrete variables (e.g. §§5.1 and 5.4) or when the number of variables within the model changes. Both of these challenges are frequently encountered when latent variables represent some unknown state of the system such as disease status or other individual classification. HMCMC efficiency is not related to whether a given problem is model or data dominant, but is very much dependent on the specific shape of the posterior itself. Often it is tested on high dimensional multivariate normal-type posterior distributions, where it is found to perform well against other approaches [3]. However, in many real-world problems, Hamiltonian trajectories can suffer from random walk-type behaviour (as was observed in example 5.3) as a result of parameter/latent variable correlations. These trajectories necessitate a large number of small intermediary steps for each MCMC update, significantly reducing algorithm performance. In contrast to PMCMC, HMCMC cannot easily be parallelized.

Finally, we come to PBPs. Generally speaking they tend to work better (in comparison to both standard approaches and HMCMC) on problems which are model dominant. The reason can be seen if we look at the limiting case in which there is no data. Here, PBPs (which are actually MBPs in this particular case) easily map out the prior distribution for model parameters (indeed equation (4.6) shows that the PBP algorithm generates random walk MH behaviour in parameter space with acceptance probability given simply by $P_{MH} = \min\{1, \pi(\theta^p)/\pi(\theta^i)\}$). The introduction of importance distributions in §4 allow for the data itself to be incorporated into the proposals, so helping maintain the efficiency of PBPs as they move out of the model-dominated regime towards the data-dominated case.

The main challenges facing PBPs are twofold: firstly, the development of fast and accurate importance distributions that help PBPs in the data dominant regime (e.g. on the right-hand side of figure 8*d*) and, secondly, the identification of PBP proposals for a broad range of distributions (i.e. extending table 1 to include other distributions). Like HMCMC, PBPs also cannot easily be parallelized. However, a potential future extension to PBPs would be to incorporate them into a particle-like framework in which multiple proposals are made and a sequentially applied particle filter is used to cull those proposals that have a low acceptance probability. This may lead to further improvements in speed under some scenarios, and will be the subject of future investigation.

One final point to mention is that while some complex models may not fit into the DAG structure required by PBPs, it may be that certain subsections of them do. Here, PBPs could still profitably be used by fixing all non-DAG elements of the model under the proposal (whereby they are incorporated into the data *y* for the purposes of the proposal step).

The introduction of PBPs offers a promising opportunity for optimizing MCMC and for further improvements, e.g. through creating particle versions of PBPs and the development and automation of efficient importance distributions under different scenarios. As such we believe PBPs are an exciting new methodology which will complement other tools in the MCMC toolbox.

Data accessibility. All of the C++ computer codes used to generate the results in §5 are included in the electronic supplementary material.

[19]Which could be measured by comparing the number of data points and the number of model parameter and latent variables.

[20]This happens in cases in which after simulating between successive time points the probability of the observed data is small, either because the observations themselves are very specific, or because multiple measurements are made.

Authors' contributions. C.M.P. introduced the ideas behind PBPs and obtained the results, S.C.B. introduced application of PBPs to quantitative genetics models, A.D.W. help clarify description of the methodology, and G.M. helped develop the manuscript along with contributing to the methodology itself. All authors gave final approval for publication.

Competing interests. We declare we have no competing interests.

Funding. This research was funded by the Strategic Research programme of the Scottish Government's Rural and Environment Science and Analytical Services Division (RESAS). A.D.W.'s contribution was funded by the BBSRC Institute Strategic Programme Grants.

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
