## [Reviewer comments · Royal Society Open Science]

Review History

RSOS-190619.R0 (Original submission)

Review form: Reviewer 1

Is the manuscript scientifically sound in its present form?

Yes

Are the interpretations and conclusions justified by the results?

No

Is the language acceptable?

Yes

Is it clear how to access all supporting data?

Yes

Do you have any ethical concerns with this paper?

No

Have you any concerns about statistical analyses in this paper?

No

Recommendation?

Major revision is needed (please make suggestions in comments)

Comments to the Author(s)

The authors present a new MCMC method, "posterior based proposals" (PBPs). The method relies on the statistical model's conditional dependence structure being able to be represented by a DAG. This structure is exploited in the ordered sampling of the latent variables in each iteration. The authors present their method, and then demonstrate speed-up in comparison with Gibb's sampling for a number of examples.

The work is interesting and worthy of publication, but I do have some concerns about the paper which I would be pleased if the authors were given an opportunity to address.

1. In the abstract, the authors claim that PBPs are competitive with HMC and PMCMC. However, they do not provide any evidence for this. In Section 7, comparisons are discussed, in particular with regards to what problems the methods are applicable to. However there are no numerical experiments to back up the pretty strong claims in the abstract. This needs to be addressed.
2. I have some concerns about the approach used to compare the performance of the methods. The authors use effective sample size per unit of CPU time, but this quantifier can give very unrepresentative values if taken from an unconverged chain, which can happen when there are strong correlations and mixing is very poor. Convergence of, for example, the KS statistic, can give a more reliable indication of the performance of MCMC algorithms.
3. My concern about the approach is that it is hard to see the method in a general setting as Table 1 appears to indicate a list of special cases rather than a unifying theory about what to do and when. I'm sure it's not the case, but it comes as rather ad hoc, and it might be instructive if more description on these derivations was given, so that the reader might be able to generalise and apply the method to a wider class of problems.
4. Similarly it is not clear from the examples how one might choose a priori which level of ID to use. Numerical results are given with ID_0, ID_1 and ID_2, but how would somebody using this algorithm know what to use?

These are my major comments, but I have also found a few typos and some more minor comments:

Pg 5 line 25, "indexes" -> "indices"

Pg 5 line 27, "known a" -> "known as"

Figure 9 is not referenced in the main text

Pg 13 line 50, "thorough" -> "through"

Review form: Reviewer 2**Is the manuscript scientifically sound in its present form?**

Yes

Are the interpretations and conclusions justified by the results?

No

Is the language acceptable?

Yes

Is it clear how to access all supporting data?

Yes

Do you have any ethical concerns with this paper?

No

Have you any concerns about statistical analyses in this paper?

Yes

Recommendation?

Reject

Comments to the Author(s)

See attached review (Appendix A).

Decision letter (RSOS-190619.R0)

23-Jul-2019

Dear Professor Pooley,

The editors assigned to your paper ("Posterior-based proposals for speeding up Markov chain Monte Carlo") have now received comments from reviewers. We would like you to revise your paper in accordance with the referee and Associate Editor suggestions which can be found below (not including confidential reports to the Editor). Please note this decision does not guarantee eventual acceptance.

Please submit a copy of your revised paper before 15-Aug-2019. Please note that the revision deadline will expire at 00.00am on this date. If we do not hear from you within this time then it will be assumed that the paper has been withdrawn. In exceptional circumstances, extensions may be possible if agreed with the Editorial Office in advance. We do not allow multiple rounds of revision so we urge you to make every effort to fully address all of the comments at this stage. If deemed necessary by the Editors, your manuscript will be sent back to one or more of the original reviewers for assessment. If the original reviewers are not available, we may invite new reviewers.

When submitting your revised manuscript, you must respond to the comments made by the

referees and upload a file "Response to Referees" in "Section 6 - File Upload". Please use this to document how you have responded to the comments, and the adjustments you have made. In order to expedite the processing of the revised manuscript, please be as specific as possible in your response.

- Data accessibility

If you wish to submit your supporting data or code to Dryad (<http://datadryad.org/>), or modify your current submission to dryad, please use the following link:
<http://datadryad.org/submit?journalID=RSOS&manu=RSOS-190619>

- Competing interests

- Authors' contributions

- Acknowledgements

- Funding statement

on behalf of Professor Len Thomas (Associate Editor) and Mark Chaplain (Subject Editor)
 openscience@royalsociety.org

Associate Editor's comments (Professor Len Thomas):

Associate Editor: 1

Comments to the Author:

We have recieved reviewers from two well-qualified scientists and i have also looked over the manuscript. While the ideas may have merit, you have failed to demonstrate that they are generally applicable or that they have performance gains over other well-known samplers such as HMC. I am recommending reubmit with major revisions. I'd like to see consideration of all of the reviewers' suggestions, including particularly a more thorough performance comaprison as suggested by the second reviewer. If your suggested method does not turn out to perform as well as competitors in many situations (or even in all), this is fine -- it is still worth documenting as an interesting idea. If it does outperform a well-implemented HMC and /or particle MCMC then all the better.

Comments to Author:

Reviewers' Comments to Author:

Reviewer: 1

Comments to the Author(s)

The authors present a new MCMC method, "posterior based proposals" (PBP). The method relies on the statistical model's conditional dependence structure being able to be represented by a DAG. This structure is exploited in the ordered sampling of the latent variables in each iteration. The authors present their method, and then demonstrate speed-up in comparison with Gibb's sampling for a number of examples.

The work is interesting and worthy of publication, but I do have some concerns about the paper which I would be pleased if the authors were given an opportunity to address.

1. In the abstract, the authors claim that PBPs are competitive with HMC and PMCMC. However, they do not provide any evidence for this. In Section 7, comparisons are discussed, in particular with regards to what problems the methods are applicable to. However there are no numerical experiments to back up the pretty strong claims in the abstract. This needs to be addressed.

2. I have some concerns about the approach used to compare the performance of the methods. The authors use effective sample size per unit of CPU time, but this quantifier can give very

unrepresentative values if taken from an unconverged chain, which can happen when there are strong correlations and mixing is very poor. Convergence of, for example, the KS statistic, can give a more reliable indication of the performance of MCMC algorithms.

3. My concern about the approach is that it is hard to see the method in a general setting as Table 1 appears to indicate a list of special cases rather than a unifying theory about what to do and when. I'm sure it's not the case, but it comes as rather ad hoc, and it might be instructive if more description on these derivations was given, so that the reader might be able to generalise and apply the method to a wider class of problems.

4. Similarly it is not clear from the examples how one might choose a priori which level of ID to use. Numerical results are given with ID_0, ID_1 and ID_2, but how would somebody using this algorithm know what to use?

These are my major comments, but I have also found a few typos and some more minor comments:

Pg 5 line 25, "indexes" -> "indices"

Pg 5 line 27, "known a" -> "known as"

Figure 9 is not referenced in the main text

Pg 13 line 50, "thorough" -> "through"

Reviewer: 2

Comments to the Author(s)

See attached review.

Author's Response to Decision Letter for (RSOS-190619.R0)

See Appendix B.

Decision letter (RSOS-190619.R1)

23-Oct-2019

Dear Professor Pooley,

I am pleased to inform you that your manuscript entitled "Posterior-based proposals for speeding up Markov chain Monte Carlo" is now accepted for publication in Royal Society Open Science.

Kind regards,
Lianne Parkhouse
Editorial Coordinator
Royal Society Open Science
openscience@royalsociety.org

on behalf of Professor Len Thomas (Associate Editor) and Professor Mark Chaplain (Subject Editor)
openscience@royalsociety.org

Associate Editor Comments to Author (Professor Len Thomas):

Thank-you for your thorough responses to the reviewers' comments, and the additional computations you've done. On this basis, I have recommended the paper be accepted.

One minor thing you may wish to change: on line 28 you say "Furthermore, PMCMC is limited to scenarios in which data is sequentially ordered (such as in time series data)." I believe the applications are much wider than this, and that this sentence could be deleted. See, for example, Section 4 of Andrieu et al. (2010) Particle Markov chain Monte Carlo Methods. JRSSA <https://doi.org/10.1111/j.1467-9868.2009.00736.x>

Appendix A

Review of ‘posterior-based proposals for speeding up MCMC’.

The authors propose a new MCMC method for fitting statistical models involving latent variables. They design a specific variant of the Metropolis–Hastings algorithm for this task. The Metropolis–Hastings algorithm requires the user to specify a proposal at each iteration, and the contribution of the present article is a method for doing this. The authors highlight some challenges of sampling in the latent variable setting, explain their ‘posterior-based proposals’, and then explore performance on a range of examples. There is some discussion of alternative approaches both at the beginning and end of the paper.

I think there are some nice ideas in the paper, but ultimately I cannot recommend it for publication at the present time. There is a vast literature on MCMC and designing proposals. For this reason, for any new method to be credible, in my opinion it should really have at least 2 of the following:

1. Be easily applicable to a broad range of models, without much user-specific input or too much tuning by hand needed
2. Have some sound theory showing that the approach is efficient in some way, e.g. the asymptotic variance of ergodic averages scales well with dimension, the resulting chain mixes geometrically quickly in many settings, or there is some other nice theory supporting its design which can be empirically verified
3. Have some strong empirical evidence showing that it is better than the state-of-the-art in many interesting settings

I think the paper makes some attempt at 1, through table 1, but ultimately doesn’t quite reach it. The proposal must be specifically designed for each model, and there are several possible options to choose from, none of which seem to be uniformly better than the others (as remarked on page 9). The authors have calculated these in some settings but I don’t think that means you can call the proposal ‘generic’ in the true sense. There are also other tuning parameters, such as U (top of page 8), for which their doesn’t seem to be a natural default choice. Some specific adjustment of proposals beyond the framework also appears to have been required in the linear mixed model example on page 13 (A^{-1} dense case) in order to get satisfactory results.

There isn’t really much in the paper on point 2, but again there is an attempt at 3. I am not convinced by this attempt, however. For all examples except the first (in which the benefit over Gibbs sampling is not very big), I would strongly argue that there are numerous other methods better positioned to be comparators than those chosen by the authors. There is discussion of Hamiltonian Monte Carlo (HMC) in the paper, but no comparisons in the experiments. Also of non-centering. I would argue that in many examples in which the authors claim to show improvements HMC with non-centering would perform extremely well, and would be the standard choice for practitioners, and I think the fair comparison would be against HMC performed in Stan [1,2] with the appropriate parametrisation for the problem chosen. I don’t agree with the discussion on page 15 about HMC, in fact some ill-conditioned distributions of the kind described by the authors are often cited as celebrated examples in which HMC performs well (e.g. [3]). There are many other celebrated strategies for MCMC and these models, such as MALA [4], and moving between centered and non-centered parametrisations within an algorithm [5], as well as other strategies beyond MCMC that work well in some of these contexts, such as INLA for mixed models [6]. There is a large literature that is not really discussed in the paper, and I think to believe that the method is credible I would need to see comparisons against at least some of these. For these reasons I cannot recommend it for publication at this time.

References

- [1] Carpenter, Bob, et al. "Stan: A probabilistic programming language." Journal of statistical

software 76.1 (2017).

[2] Betancourt, Michael, and Mark Girolami. "Hamiltonian Monte Carlo for hierarchical models." *Current trends in Bayesian methodology with applications* 79 (2015): 30.

[3] Chen, Lingyu, Zhaohui Qin, and Jun S. Liu. "Exploring hybrid monte carlo in bayesian computation." *sigma* 2 (2001): 2-5.

[4] Roberts, Gareth O., and Richard L. Tweedie. "Exponential convergence of Langevin distributions and their discrete approximations." *Bernoulli* 2.4 (1996): 341-363.

[5] Yu, Yaming, and Xiao-Li Meng. "To center or not to center: That is not the question—an Ancillarity–Sufficiency Interweaving Strategy (ASIS) for boosting MCMC efficiency." *Journal of Computational and Graphical Statistics* 20.3 (2011): 531-570.

[6] Fong, Youyi, Håvard Rue, and Jon Wakefield. "Bayesian inference for generalized linear mixed models." *Biostatistics* 11.3 (2010): 397-412.

Appendix B

Dear Editor,

We thank the Associate Editor and reviewers for their insightful comments. As outlined in the detailed response letter below, we have taken them all carefully on board and revised the manuscript accordingly. We believe the changes made to the paper make it much stronger.

Below we outline our responses to the specific issues raised (these changes have been highlighted in the main text):

Associate Editor's comments (Professor Len Thomas):

Associate Editor: 1

Comments to the Author:

We have received reviewers from two well-qualified scientists and i have also looked over the manuscript. While the ideas may have merit, you have failed to demonstrate that they are generally applicable or that they have performance gains over other well-known samplers such as HMC. I am recommending resubmit with major revisions. I'd like to see consideration of all of the reviewers' suggestions, including particularly a more thorough performance comparison as suggested by the second reviewer. If your suggested method does not turn out to perform as well as competitors in many situations (or even in all), this is fine -- it is still worth documenting as an interesting idea. If it does outperform a well-implemented HMC and /or particle MCMC then all the better.

A thorough comparison against optimised HMCMC and PMCMC methods, as well as non-centred parameterisations, has now been made. The results are presented in section 5, and a detailed description of these methods is given in three new Appendices I-K. The results highlight that no one method is best in all scenarios, but that PBPs either achieve near optimal performance or are considerably faster than any of the other methods tried (by up to a factor of 10).

Regarding generality, it is fair to say that PBPs are restricted to models that make use of distributions for which appropriate proposals can be made (Table 1 shows examples for the most commonly used distributions, but doubtless others can be found). However, these do cover a very large proportion of statistical models in the literature. To emphasise this, the following text has been added to the beginning of Section 2:

“PBPs are potentially applicable to any statistical model whose conditional dependence structure can be represented by a Direct Acyclic Graph (DAG) [18], as illustrated in Fig. 2. This encompasses a vast range of statistical models including mixed models [19], generalised linear mixed models [20], hidden Markov models [21], discrete time Markov processes [22], and most of the models that can be defined in automated Bayesian software, such as WinBUGS [23], JAGS [24] or Stan [25], which specifically assume a DAG structure.”

Furthermore, we have added the following text to the end of the discussion:

“The introduction of PBPs offers a promising opportunity for optimising MCMC and for further improvements, *e.g.* through creating particle versions of PBPs and the development and automation of efficient importance distributions under different scenarios. As such we believe PBPs are an exciting new methodology which will complement other tools in the MCMC toolbox.”

Comments to Author:

Reviewers' Comments to Author:

Reviewer: 1

Comments to the Author(s)

The authors present a new MCMC method, "posterior based proposals" (PBPs). The method relies on the statistical model's conditional dependence structure being able to be represented by a DAG. This structure is exploited in the ordered sampling of the latent variables in each iteration. The authors present their method, and then demonstrate speed-up in comparison with Gibb's sampling for a number of examples.

The work is interesting and worthy of publication, but I do have some concerns about the paper which I would be pleased if the authors were given an opportunity to address.

1. In the abstract, the authors claim that PBPs are competitive with HMC and PMCMC. However, they do not provide any evidence for this. In Section 7, comparisons are discussed, in particular with regards to what problems the methods are applicable to. However there are no numerical experiments to back up the pretty strong claims in the abstract. This needs to be addressed.

These concerns have now been thoroughly addressed by a major revision of the entire of Section 5, which now includes a comparison of HMCMC and PMCMC results (where possible) with PBPs. Results of this can be found in Figs. 5 – 8 for the different examples, and these demonstrate that whilst each method has its strengths, PBPs exhibit a substantial increase in computational speed against these state-of-the-art approaches in some cases and across all examples considered are competitive with them.

2. I have some concerns about the approach used to compare the performance of the methods. The authors use effective sample size per unit of CPU time, but this quantifier can give very unrepresentative values if taken from an unconverged chain, which can happen when there are strong correlations and mixing is very poor. Convergence of, for example, the KS statistic, can give a more reliable indication of the performance of MCMC algorithms.

The effective sample size per unit of CPU time was used because this has become a standard way to measure performance, and so will be familiar to most of the readers of the paper. The KS statistic could doubtless have also be used, but here care was taken to make sure that all the result are based on well-mixed MCMC chains (specifically an effective sample size typically exceeding 1000 and at least greater than 500). To emphasise this, footnote 14 has been added:

“Note, all MCMC chains were run long enough to be well mixed, with ESS typically exceeding 1000 and at least greater than 500.”

3. My concern about the approach is that it is hard to see the method in a general setting as Table 1 appears to indicate a list of special cases rather than a unifying theory about what to do and when. I'm sure it's not the case, but it comes as rather ad hoc, and it might be instructive if more description on these derivations was given, so that the reader might be able to generalise and apply the method to a wider class of problems.

Table 1 does provide the PBP formulae required for many widely used distributions that underpin the vast majority of statistical models covered in the literature. However, we have yet to identify an

algorithmic approach to the development of PBP for an arbitrary distribution and this remains the subject of further research. For readers interested in pursuing this goal we describe in detail the derivation of the form of the PBP required for Poisson and Normal distributions in Appendix B.

To clarify these points we have added the following sentences at the end of Section 3.3:

“Deriving sampling schemes which also satisfy condition 2 is a non-trivial task guided by intuition and trial and error. Extension of Table 1 to encompass a more comprehensive list of possible sampling distributions will be the subject of future research. The validity of Eqs.(6) for both Poisson and normal IDs is explicitly demonstrated in Appendix B.”

4. Similarly it is not clear from the examples how one might choose a priori which level of ID to use. Numerical results are given with ID_0, ID_1 and ID_2, but how would somebody using this algorithm know what to use?

To address this we have modified the following paragraph at the end of Section 4:

“Choosing which level of ID to optimise PBPs involves a trade-off between the computational cost of generating IDs with the size of posterior jumps (and hence improvement in mixing) they allow. Unfortunately determining *a priori* which option is best is challenging. Indeed, in the results section below we find examples for which ID_0 , ID_1 and ID_2 each represent optimum solutions for different problems. From the point of view of the user, the pragmatic approach to take is to first try ID_0 (which is the easiest to implement) and if that doesn't help mixing then try ID_1 and so on and so forth. Identification of optimal IDs will be the subject of active future research.”

These are my major comments, but I have also found a few typos and some more minor comments:

Pg 5 line 25, "indexes" -> "indices"

Pg 5 line 27, "known a" -> ""known as"

Figure 9 is not referenced in the main text

Pg 13 line 50, "thorough" -> "through"

We thank the reviewer for pointing out these typos, which have now been corrected.

Reviewer: 2

Review of ‘posterior-based proposals for speeding up MCMC’.

The authors propose a new MCMC method for fitting statistical models involving latent variables. They design a specific variant of the Metropolis–Hastings algorithm for this task. The Metropolis–Hastings algorithm requires the user to specify a proposal at each iteration, and the contribution of the present article is a method for doing this. The authors highlight some challenges of sampling in the latent variable setting, explain their ‘posterior-based proposals’, and then explore performance on a range of examples. There is some discussion of alternative approaches both at the beginning and end of the paper.

I think there are some nice ideas in the paper, but ultimately I cannot recommend it for publication at the present time. There is a vast literature on MCMC and designing proposals. For

this reason, for any new method to be credible, in my opinion it should really have at least 2 of the following:

1. Be easily applicable to a broad range of models, without much user-specific input or too much tuning by hand needed

The vast majority of models make use of the distributions in Table 1, so we believe PBPs are applicable to a broad range of models (see description in Section 2, as mentioned above).

Regarding user input, a distinction needs to be made between MBPs and PBPs. For MBPs there is essentially no hand-tuning required. There is a one-to-one correspondence between the model distributions and the proposals (as described in Table 1). Such a process could be easily automated. The following paragraph has been added to the top of page 10:

“One of the desirable features of MBPs is that they require no hand-tuning. There is a one-to-one correspondence between the model distributions and the proposals, as outlined in Table 1, and so they can be implemented in an automated manner. However in cases in which data substantially restricts model parameters and latent variables, higher order importance distributions become necessary.”

For PBPs, however, the user does need to work out appropriate importance distributions, and this requires some careful consideration. As mentioned above, automated methods for doing this will be the subject of future research.

A summary of these issues and a comparison with other approaches is now given in Table 2, along with a discussion in Section 7.

2. Have some sound theory showing that the approach is efficient in some way, e.g. the asymptotic variance of ergodic averages scales well with dimension, the resulting chain mixes geometrically quickly in many settings, or there is some other nice theory supporting it's design which can be empirically verified

The theoretical reason for it being faster lies in the fact that for PBPs, random walk behaviour occurs in parameter space but not in latent variable space (just as for PMCMC). To clarify this the following sentence has been added to Section 3.4:

“The algorithm above performs a random walk through the parameter space defined by the posterior, but because the dimensionality of θ is typically much less than ξ , it is expected to mix at a much faster rate⁸ than standard MCMC, which performs a random walk in both θ and ξ .”

with the footnote:

⁸ Subject to sufficiently large jumping size j in Eq.(8) being possible.”

3. Have some strong empirical evidence showing that it is better than the state-of-the-art in many interesting settings

Section 5 now includes results from tuned HMCMC and PMCMC algorithms, as well as using non-centred parameterisation. These demonstrate that not only do these other methods all have regimes in which they perform poorly (or are not even applicable), but that the PBP results are either near to the fastest or are substantially faster than any other approach. This has been clarified in the abstract.

I think the paper makes some attempt at 1, through table 1, but ultimately doesn't quite reach it.

The proposal must be specifically designed for each model, and there are several possible options to choose from, none of which seem to be uniformly better than the others (as remarked on page 9). The authors have calculated these in some settings but I don't think that means you can call the proposal 'generic' in the true sense.

There are also other tuning parameters, such as U (top of page 8), for which their doesn't seem to be a natural default choice.

The following sentence has been included at the end of section 3.4:

"Appendix H shows that mixing is not sensitive to the exact value of U , and is approximately optimised when $U=4$ (as subsequently used)¹⁰."

With footnote:

¹⁰ In practice the optimum U will depend on the relative CPU time needed for PBP and standard updates. If standard updates are much slower it makes sense for U to be higher, but typically they are of a similar speed."

Furthermore, it can be pointed out that nearly all of the methods tried (with the exception of the standard Gibbs approach) require some level of tuning for optimal performance, and PBPs are no different in this respect.

Some specific adjustment of proposals beyond the framework also appears to have been required in the linear mixed model example on page 13 (A-1 dense case) in order to get satisfactory results.

This example no longer appears in the paper. However, because the results for the non-centred approach are good, the following paragraph has been added to the end of section 5.3:

"PBP and NCP approaches have been applied to mixed models and generalised mixed models (binary disease data) with diagonal, sparse and dense A (results not shown) and overall PBPs were not found to perform substantially faster than NCP, and in some cases were slower. Consequently, for these types of model, standard approaches using NCP may prove to be the best method to use, particularly given the relative ease with which they can be implemented (Table 2)."

There isn't really much in the paper on point 2, but again there is an attempt at 3. I am not convinced by this attempt, however. For all examples except the first (in which the benefit over Gibbs sampling is not very big), I would strongly argue that there are numerous other methods better positioned to be comparators than those chosen by the authors. There is discussion of Hamiltonian Monte Carlo (HMC) in the paper, but no comparisons in the experiments. Also of non-centering. I would argue that in many examples in which the authors claim to show improvements HMC with non-centering would perform extremely well, and would be the standard choice for practitioners, and I think the fair comparison would be against HMC performed in Stan [1,2] with the appropriate parametrisation for the problem chosen.

See above for a discussion giving comparisons to other methods. With regards to comparing PBPs with HMCMC, we took the decision to write efficient C++ code (available in the supplementary material) from scratch separately for each the models (just as with the other algorithms), rather than make a direct comparison with Stan. This allows us to perform a well-controlled, fair comparison between all the methods. It's true to say that Stan does implement NUTS, but this is not expected to be any faster than the well-tuned standard approach we use here [1]. We have included the following sentence at the end of Appendix J:

“This paper takes a brute force approach to find the optimal HMC implementation used to compare with other methods. For each set of simulated data inference is carried out using a large number of different values of integral length L_e , and the most efficient of these is used to construct the HMCMC curves in Figs. 6(d) and 7(b). An example of this process is shown in Fig. J, which demonstrates how the NCP HMCMC results in Fig. 7(b) were generated. Note, under realistic models the optimal results from NUTs are found to have very similar computational efficiency to HMCMC tuned in this fashion [9].”

I don't agree with the discussion on page 15 about HMC, in fact some ill conditioned distributions of the kind described by the authors are often cited as celebrated examples in which HMC performs well (e.g. [3]).

In light of the results from the HMCMC method, the discussion has largely been changed.

There are many other celebrated strategies for MCMC and these models, such as MALA [4],

As now pointed out at the beginning of section 5:

“The Metropolis-adjusted Langevin algorithm (MALA) is a special case of HMCMC in which only a single step is taken.”

and moving between centered and non-centered parametrisations within an algorithm [5]

The following comment has been added at the beginning of section 5:

“More complicated schemes which make use of partial CP/NCP proposals and interweaving different parameterisations [2-5] are not considered here.”

, as well as other strategies beyond MCMC that work well in some of these contexts, such as INLA for mixed models [6].

INLA, whilst fast, is an approximate method, so we don't feel it naturally fits into this paper.

There is a large literature that is not really discussed in the paper, and I think to believe that the method is credible I would need to see comparisons against at least some of these.

This has now been addressed.

References

- [1] M. D. Hoffman and A. Gelman, "The No-U-Turn sampler: adaptively setting path lengths in Hamiltonian Monte Carlo," *Journal of Machine Learning Research*, vol. 15, no. 1, pp. 1593-1623, 2014.
- [2] P. Neal and G. Roberts, "A case study in non-centering for data augmentation: Stochastic epidemics," *Stat. Comput.*, vol. 15, no. 4, pp. 315–327, 2005.
- [3] O. Papaspiliopoulos, G. O. Roberts, and M. Skold, "A general framework for the parametrization of hierarchical models," (in English), *Statistical Science*, vol. 22, no. 1, pp. 59-73, Feb 2007.
- [4] O. Papaspiliopoulos, G. O. Roberts, and M. Skold, "Non-centered parameterisations for hierarchical models and data augmentation," in *Bayesian Statistics 7*, 2003, vol. 307, pp. 307-326: Oxford University Press, USA.
- [5] Y. Yu and X.-L. Meng, "To center or not to center: That is not the question—an Ancillarity–Sufficiency Interweaving Strategy (ASIS) for boosting MCMC efficiency," *Journal of Computational and Graphical Statistics*, vol. 20, no. 3, pp. 531-570, 2011.